# Chromatin accessibility of circulating CD8+ T cells predicts treatment response to PD-1 blockade in patients with gastric cancer

Hyun Mu Shin [1,2,3,12✉], Gwanghun Kim[2,3,4,12], Sangjib Kim [5,12], Ji Hyun Sim[4,12], Jiyeob Choi[2,6], Minji Kim[2,3,4], Minsuk Kwon [7], Sang-Kyu Ye[2,3,8,9], Dong-Sup Lee[1,2,3,4,9], Seung Woo Cho [10], Seung Tae Kim[7], Jeeyun Lee [7,11✉] & Hang-Rae Kim [1,2,3,4,9✉]

Although tumor genomic profiling has identified small subsets of gastric cancer (GC) patients with clinical benefit from anti-PD-1 treatment, not all responses can be explained by tumor sequencing alone. We investigate epigenetic elements responsible for the differential response to anti-PD-1 therapy by quantitatively assessing the genome-wide chromatin accessibility of circulating CD8+ T cells in patients' peripheral blood. Using an assay for transposase-accessible chromatin using sequencing (ATAC-seq), we identify unique open regions of chromatin that significantly distinguish anti-PD-1 therapy responders from non-responders. GC patients with high chromatin openness of circulating CD8+ T cells are significantly enriched in the responder group. Concordantly, patients with high chromatin openness at specific genomic positions of their circulating CD8+ T cells demonstrate significantly better survival than those with closed chromatin. Here we reveal that epigenetic characteristics of baseline CD8+ T cells can be used to identify metastatic GC patients who may benefit from anti-PD-1 therapy.

[1] Wide River Institute of Immunology, Seoul National University, Hongcheon, Republic of Korea. [2] Department of Biomedical Sciences, Seoul National University College of Medicine, Seoul, Republic of Korea. [3] BK21 FOUR Biomedical Science Project, Seoul National University College of Medicine, Seoul, Republic of Korea. [4] Department of Anatomy and Cell Biology, Seoul National University College of Medicine, Seoul, Republic of Korea. [5] Department of Mathematics, College of Science, Korea University, Seoul, Republic of Korea. [6] Department of Preventive Medicine, Seoul National University College of Medicine, Seoul, Republic of Korea. [7] Department of Medicine, Division of Hematology-Oncology, Samsung Medical Center, Sungkyunkwan University School of Medicine, Seoul, Republic of Korea. [8] Department of Pharmacology, Seoul National University College of Medicine, Seoul, Republic of Korea. [9] Medical Research Institute, Seoul National University College of Medicine, Seoul, Republic of Korea. [10] Department of Biomedical Engineering, School of Life Sciences, Ulsan National Institute of Science and Technology (UNIST), Ulsan, Republic of Korea. [11] Department of Intelligent Precision Healthcare Convergence, Sungkyunkwan University, Suwon, Korea. [12] These authors contributed equally: Hyun Mu Shin, Gwanghun Kim, Sangjib Kim, Ji Hyun Sim. ✉email: hyunmu. shin@snu.ac.kr; jyunlee@skku.edu; hangrae2@snu.ac.kr

Studies of immune cell exhaustion and tumor-infiltrating lymphocyte (TIL) dysfunction due to persistent antigenic stimuli (e.g., chronic viral infection and tumors) have revealed cell surface proteins that are involved in the functional impairment of immune cells; these proteins include multiple co-inhibitory or checkpoint receptors, such as cytotoxic T lympho-cyte antigen-4 (CTLA-4, CD152), programmed death-1 (PD-1, CD279), T cell immunoglobulin and mucin-domain containing-3 (TIM-3), and lymphocyte-activation gene (LAG-3)[1]. Conse-quently, clinical trials involving in vivo administration of neu-tralizing antibodies (Abs) against immune checkpoints (i.e., immune checkpoint inhibitors [ICIs]) have been conducted to functionally restore T cells[1,2]. These ICI drugs are currently approved for those patients who have failed at least two prior lines of systemic chemotherapy in metastatic gastric cancer (mGC)[3,4]. Pembrolizumab has been provisionally approved based on biomarker selection (PD-L1 immunohistochemistry [IHC] status) while nivolumab has been approved without biomarker selection. We recently reported on phase II study of single agent pembrolizumab in mGC which identified remarkable and durable response in the microsatellite instability-high (MSI-H) and Epstein–Barr virus (EBV) positive GC subtypes[5]. However, the incidence of MSI-H and EBV positivity in mGC is relatively low in the range between 2 to 4%. Hence, we need to understand who may potentially benefit from ICI beyond MSI-H and EBV posi-tivity in their tumor characteristics in order to increase the rate of GC patients with clinical benefit.

Recent studies have mainly focused on tumor genomics and PD-L1 scores as predictors for response to ICIs[6,7]. Although an in-depth understanding of the regulation of tumor–host inter-actions has been underscored to optimize treatment strategies in immunotherapy, the role of circulating CD8+ T cells have not been extensively investigated in GC patients, especially in corre-lation to treatment response. In the present study, we investigate the correlation between changes in circulating CD8+ T cells (baseline and 3 to 6 weeks pembrolizumab) and response to anti-PD-1 treatment using flow cytometric analysis in our previous phase II pembrolizumab cohort[5]. Next, we analyze the baseline CD8+ T cells using an assay for transposase-accessible chromatin using sequencing (ATAC-seq)[8] to further characterize the epi-genetic landscape of circulating CD8+ T cells which may reflect host-tumor interactions. Chronic stress[9–11] caused by factors such as pathogens, microbiota, acidity, hypoxia, glucose depri-vation, and the tumor microenvironment can cause epigenetic changes[12,13] that lead to subtle to profound alterations in the immune system[14,15]. These host-tumor interactions can alter epigenetic characteristics of circulating CD8+ T cells which may increase or decrease response to anti-PD-1 treatment in cancer patients[16,17].

Our analysis show that chromatin openness in baseline circu-lating CD8+ T cells distinguish responder group (R) from non-responder group (NR) in the discovery cohort and the validation cohort. All EBV-positive patients and MSI-H patients who responded to pembrolizumab have high openness of chromatin at specific genomic positions of baseline CD8+ T cells. Furthermore, mGC patients with high openness chromatin of CD8+ T cells show longer PFS following anti-PD-1 treatment when compared to those with closed chromatin. These results present that openness of chromatin at specific genomic positions of baseline circulating CD8+ T cells can predict clinical outcome to treat-ment response following anti-PD-1 treatment.

## Results
### Clinical characteristics of patients with gastric cancer. For discovery cohort, cryopreserved blood lymphocyte samples from

mGC patients participating in a phase II pembrolizumab monotherapy clinical trial (clinicaltrials.gov identifier NCT#02589496)[5] were used in this study. Of the 61 patients enrolled, only viable blood lymphocyte samples from 32 patients had baseline, 1, 2 cycles post-pembrolizumab blood lymphocytes collected were used for further analysis. For validation cohort, 52 mGC patients who received pembrolizumab for third-line treat-ment were consented for baseline blood collection. Clinical characteristics of all patients with mGC are provided in Supple-mentary Tables 1 and 2. In total, 84 samples were analyzed with ATAC sequencing. Of 84 patients, 11 patients were MSI-H GC patients (8 responders) and 10 patients were EBV-positive GC (5 responders). In an entire cohort, 28 of 84 patients (33.3%) were responders to anti-PD-1 therapy; 15 of 28 (53.6%) responders were non-EBV or MSS patients. Taken together, 53.6% of the patients lacked of known predictive factors such as MSI-H or EBV-positive in their tumor specimen but still responded to anti-PD-1 therapy. PD-L1 IHC (immunohistochemistry) score using 22C3 antibody is provided in all available tumor specimen (Supplementary Tables 1 and 2).

In the flow cytometric analysis for discovery cohort, there was no significant difference in the frequencies of CD8+ T cells and PD-1+CD8+ T cells in responders (complete response [CR], partial response [PR]) or non-responders (stable disease [SD] or progressive disease [PD]) to pembrolizumab (Fig. 1a–b and Supplementary Fig. 1a). In flow cytometric analysis of pre- and post-treatment (3 to 6 weeks) samples, the proportion of Ki-67+ CD8+ T cells (% proliferating cells [Ki-67+], baseline 3.42 ± 0.33, post-treatment 6.72 ± 1.15, mean ± standard error of the mean [SEM], $P = 0.008$, Wilcoxon signed rank test) increased after pembrolizumab (Fig. 1c, left panel). Further, the frequency (%) of proliferating PD-1+CD8+ T cells (baseline 5.22 ± 0.76, post-treatment 14.4 ± 1.78, mean ± SEM, $P < 0.0001$, Wilcoxon signed rank test) increased after pembrolizumab (Fig. 1c, middle panel). However, there was no substantial difference in percentage of Ki-67+ in PD-1+CD8+ T cells among responders and non-responders (Fig. 1c, right panel); this finding is consistent with the results of previous studies[18,19].

**Normalization process of ATAC-seq data among samples.** CD8+ T cells isolated from baseline blood samples of patients enrolled in a phase II pembrolizumab trial[5] were subjected to genome-wide chromatin assay for ATAC-seq (Fig. 2a). We developed a normalization method to quantitatively compare the ATAC-seq data. Given the increased sensitivity, reproducibility, and large dynamic range of ATAC technology, the necessity of normalizing these large ATAC-seq datasets exceeded the requirements of ATAC library and deep-sequencing quality guidelines (Supplementary Table 3)[20,21]. We applied a normal-ization process developed for massive computation loads[22,23]. We performed two critical selection steps to discover distinctive peaks differentiating responders from non-responders within our ana-lysis of human genome (hg19)-mapped ATAC-seq data (Fig. 2b). The first step was the selection of a universal control to identify consensus peaks to adopt as normalization controls (Fig. 2b–d). We matched 4,798 consensus ATAC peaks among previously studied CD8+ T-cell subsets[17] with DNase I hypersensitive (DH) uniform peaks from the ENCODE project[24] located on the TSS or 5′-untranslated regions (UTR), which are transcriptionally active regions of the genome. Among these, 533 consensus ATAC peaks appeared in the ATAC-seq CD8+ T-cell data used in this study. To obtain more accurate normalization controls from selected consensus ATAC peaks, we adopted a strategy of selecting peaks with low coefficients of variation (CVs), resulting in high simi-larity due to low variability among ATAC peaks for the samples.

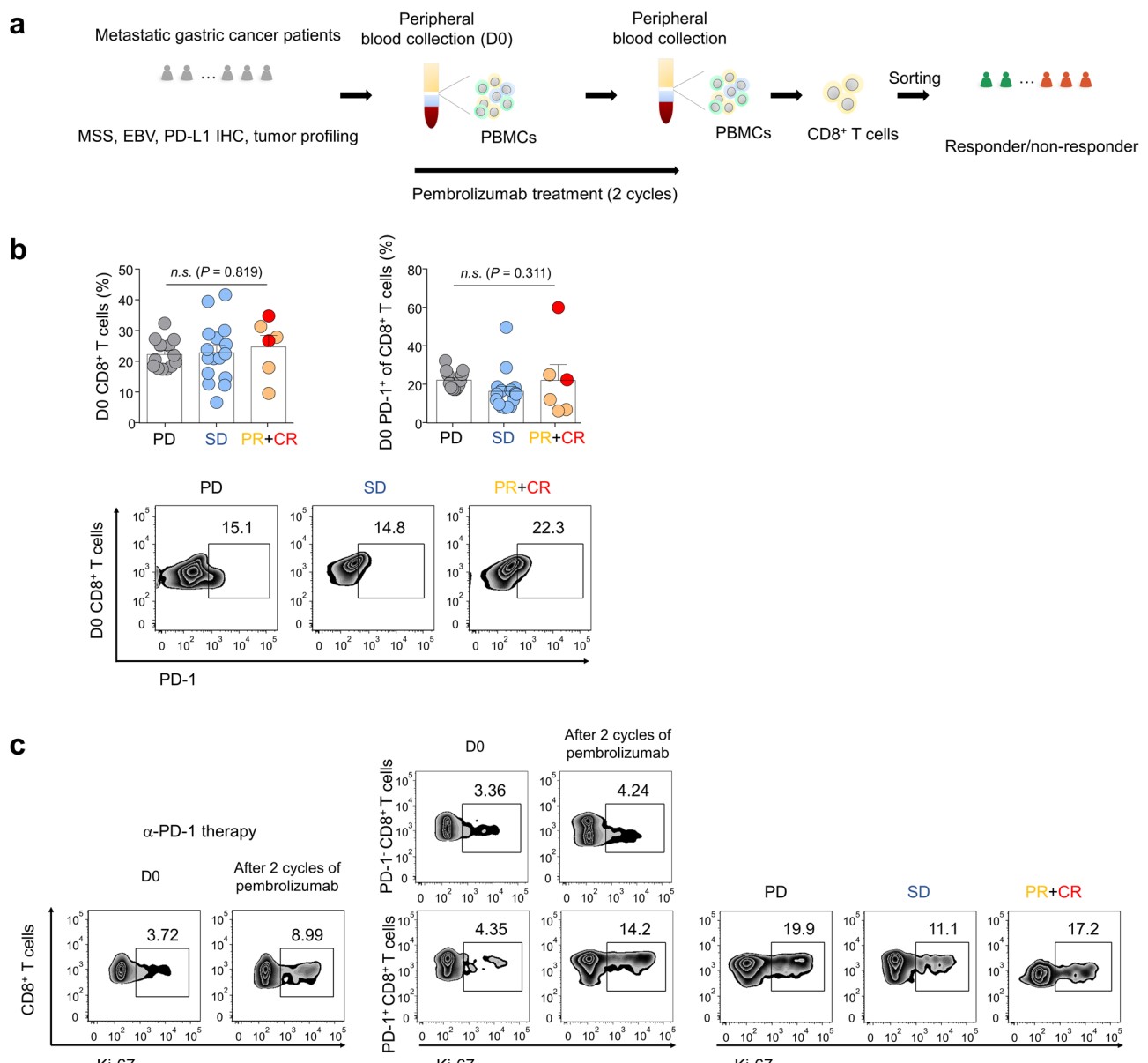

**Fig. 1 PD-1+CD8+ T cell proliferation after pembrolizumab treatment. a** Outline of flow cytometric analysis and transposase-accessible chromatin using sequencing (ATAC-seq)-based molecular diagnosis. Peripheral blood mononuclear cells (PBMCs) from patients with metastatic gastric cancer (mGC) were stained with antibodies (Abs) to CD8, programmed death-1 (PD-1), and Ki-67 at baseline and after 2 cycles of pembrolizumab treatment. MSS: microsatellite stability; EBV: Epstein–Barr virus; PD-L1: programmed death-ligand 1; IHC: immunohistochemistry. **b** Frequency of CD8+ T cells and PD-1+ CD8+ T cells in patients with mGC at baseline (GC PD, $n = 13$ patients; SD, $n = 16$; PR, $n = 4$; CR, $n = 2$). Representative zebra plots showing the frequency of PD-1+CD8+ T cells. Bars represent as means + SEM, Data were compared using the one-way ANOVA test (left graph: $P = 0.819$, $F(2, 32) = 0.2008$; right graph: $P = 0.311$, $F(2, 32) = 1.213$). n.s., not significant. PD: progression disease; SD: stable disease; PR: partial response; CR: complete response. **c** Representative zebra plots showing the Ki-67 expression of CD8+ T cells and CD8+ T-cell subsets (center), PD-1-CD8+ and PD-1+CD8+ T cells, in GC patient at baseline and pembrolizumab treatment. Representative zebra plots showing the Ki-67 expression of PD-1+CD8+ T cells in PD, SD, PR + CR groups.

Thus, we selected 232 ATAC peaks, of which 20 (Supplementary Fig. 2 and Supplementary Data 1) were highly evolutionarily conserved among 17 species (Fig. 2d) and could be useful for epigenomic studies with other cell types (Supplementary Fig. 3a–c). Finally, we applied these peaks for normalization within a cohort. We defined the normalization factor ($F$) of each sample as the mean height ($h$) of all samples in a cohort ($C$) divided by the height of a corresponding sample (Supplementary Fig. 4). The use of a control set to quantify and compare ATAC-seq data could enhance the applicability of ATAC-seq data.

**Epigenetic signatures for pembrolizumab treatment**. Next, we identified differential peaks of chromatin regions in the ATAC-seq between responders and non-responders to pembrolizumab. Briefly, peaks in the ATAC-seq reflect the "openness" in chromatin structure which can be influenced by tumor-host interactions. We identified 2,560 differential peaks using three peak callers[25–27] using peak-calling packages (Supplementary Table 4 and Fig. 3a). The area ($A$) values of 2,560 differential peaks in each sample were then normalized using the $F$ value determined in the first step, allowing quantitative comparisons among

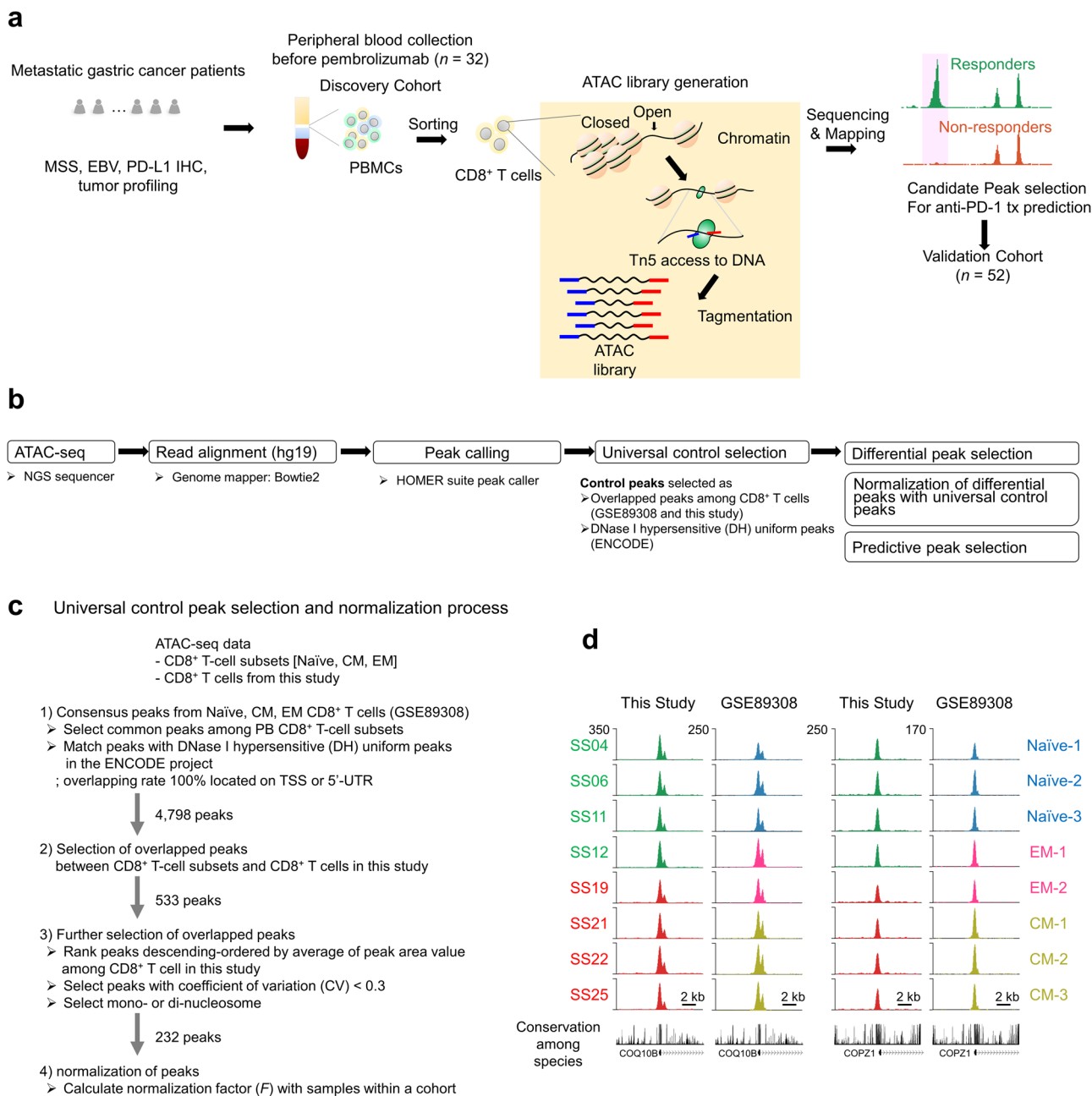

samples. The $F$ value of each sample (Supplementary Fig. 5a) and the corresponding area value (Supplementary Fig. 5b) were stabilized by the number of controls. ATAC-seq data for 2,560 differential peaks were consistently normalized with $F$ values calculated using three sets of controls, ranked −5, −20, and −50 (Supplementary Table 5) to obtain a normalized area value ($F \times A$). We used these normalized area values to further select differential peaks (Fig. 3a) as followed: 68, 121 and 149 peaks by three sets of controls respectively satisfied the criteria of average area value of peaks > average area value of total peaks and $P <$ 0.05 (Mann–Whitney $U$ test). Of these peaks, 67 shared by three sets of controls were rank-ordered by differences in the average values (Supplementary Fig. 6). For example, a control set of 20 reduced the difference in overall peak increases (corresponding to area values) among samples, allowing quantitative sample comparison (Supplementary Fig. 7). Among the 67 differential peaks (Supplementary Data 2), we visualized nine representative peaks

using the University of California, Santa Cruz (UCSC) genome browser (Fig. 3b); these peaks were located near active regulatory elements[28], suggesting that they may be regulated by various transcription factors (defined in the ENCODE projects and Supplementary Fig. 8). To explore the predictability of the selected peaks as diagnostic targets, we performed receiver operating characteristic (ROC) analysis of these targets (Fig. 3c). The nine representative targets showed statistically fair diagnostic ability[29] with the area under the ROC curve (AUROC) ≥ 0.7 and target sensitivity and specificity of 88.0 ± 3.6% and 70.0 ± 3.9% (mean ± SEM), respectively (Fig. 3d).

**Chromatin openness to response to pembrolizumab treatment.** Chromatin openness of differential peaks in CD8$^+$ T cells distinguished between a responder group (R) and a non-responder group (NR) in the discovery (discovery cohort, $n = 32$) (Fig. 4a and Supplementary Table 6). All EBV-positive patients and MSI-

**Fig. 2 Outline of assay for transposase-accessible chromatin using sequencing (ATAC-seq)-based molecular diagnosis and peak selection processes.**
**a** To identify biomarkers that can predict the benefits of anti-PD-1 therapy, we processed PBMCs from patients with mGC by generating an ATAC library and performing deep sequencing. The transposition reaction resulted in fragmented genomic DNA; transposons were inserted by Tn5 transposases into open and accessible chromatins. Following ATAC library sequencing and mapping to the human genome, genome-wide analysis revealed differences between responders and non-responders to anti-PD-1 therapy. MSS: microsatellite stability; EBV: Epstein–Barr virus; PD-L1: programmed death-ligand 1; IHC: immunohistochemistry. **b** After ATAC library generation, each library pool was quantified using a Bioanalyzer and sequenced on a single lane of the NextSeq 500 system using 75-bp single reads. Sequencing read (fastq) files were mapped using the human genome database (hg19) and Bowtie v. 2.0 software. Further analysis was performed in two stages: (1) control peak selection to identify consensus peaks among CD8$^+$ T-cell subsets known as DNase I hypersensitive (DH) sites, followed by normalization of peaks from each patient using control peaks; and (2) predictive peak selection to distinguish response and non-response to anti-PD-1 therapy after normalization of the selected differential peaks. NGS: next generation sequencing; hg: human genome. **c** Universal control peak selection and normalization. After peak calling, universal control selection was performed to identify consensus peaks as normalization controls. In total, 4,798 consensus assays were matched for transposase-accessible chromatin (ATAC) peaks among previously studied CD8$^+$ T-cell subsets with DNase I hypersensitive (DH) uniform peaks from the ENCODE project; these peaks were located on transcriptional start site (TSS) or 5′-untranslated regions (UTR). In addition, 533 consensus ATAC peaks were detected in ATAC-seq CD8$^+$ T-cell data used in this study. To identify accurate normalization controls among selected consensus ATAC peaks, the coefficient of variation (CV; i.e., the ratio of the standard deviation [SD] to the mean) was used to define the extent of ATAC peak variability among samples and peaks with mono- and di-nucleosome patterns. First, the area of the peak was calculated using the area of base pairs within the range of the peak; these areas were used to calculate CV values for each peak. In total, 232 ATAC peaks were detected which satisfied two criteria: low variance (CV < 0.3) and uniqueness of peak width (< 500 bp). Notably, the top 20 peaks in terms of average peak area were highly evolutionarily conserved among 17 species; these peaks, which could be useful for genomic studies of other species, were used to calculate the within-cohort normalization factor (F). PB: peripheral blood; EM: effector memory; CM: central memory. **d** Two representative control peaks shown in a genome browser. Conservation levels among 17 species are shown at the bottom of the corresponding peak. The y-axis shows read counts.

H patients who responded to pembrolizumab had high openness of chromatin at specific genomic positions of baseline CD8$^+$ T cells (Fig. 4a). One MSI-H patient (SS20) with de novo resistance to pembrolizumab had closed chromatin at the corresponding genomic positions of baseline CD8$^+$ T cells (Fig. 4a). In all, patients with high chromatin openness of baseline circulating CD8$^+$ T cells were enriched in responder group when compared to non-responder group with sensitivity of 87.8 ± 3.6% (mean ± SEM) and specificity of 69.7 ± 3.6% (Fig. 4a, Supplementary Table 6 and Supplementary Fig. 9). Nine regions with opened chromatin were sorted in descending order of accuracy values (Supplementary Table 7); this order was used to convert ATAC-seq data openness values to a weighted score (see Methods). When all 9 significant peaks were summated, sensitivity was 100.0% and specificity was 90.9% for distinguishing R from NR to pembrolizumab (Fig. 4b, c). In the discovery set, patients with high chromatin openness in baseline circulating CD8$^+$ T cells ($n = 12$, 12 of 32, 37.5%) (Fig. 4b) demonstrated longer median PFS (not reached) than those with closed chromatin at the corresponding genomic position of circulating CD8$^+$ T cells (median PFS, range, 2.7 months; $n = 20$, 62.5% of patients) (Fig. 4d).

**Evaluation of chromatin openness to predict anti-PD-1 therapy.** In the validation cohort ($n = 52$), patients with high chromatin openness of baseline circulating CD8$^+$ T cells were enriched in responder group when compared to non-responder group regardless (Fig. 5a, Supplementary Table 8 and Supplementary Fig. 10). All MSI-H ($n = 4$) and EBV-positive ($n = 2$) patients who responded to pembrolizumab had high chromatin openness. SS007 patient with EBV-positive GC who had closed chromatin openness did not respond to anti-PD-1 treatment. Of 13 non-EBV, MSS patients who responded, 11 (86%) patients had high chromatin openness. Four of the nine selected chromatin "open" regions showed statistically significant differences in PFS in the validation cohort, but nonetheless, each median PFS was greater in patients with high chromatin openness than in patients with closed chromatin openness. The nine targets showed reasonable discriminative ability (AUROC > 0.544) and target sensitivity and specificity of 80.2 ± 3.5% and 46.7 ± 3.9% (mean ± SEM), respectively (Supplementary Table 8 and Supplementary Fig. 10). The combination of these peaks enhanced the discriminative ability in the validation

cohort (AUROC 0.717) and sensitivity and specificity of this combination was reached to 88.9% and 58.8%, respectively (Fig. 5b). Accordingly, the combination of all 9 peaks predicted response to pembrolizumab very well (Fig. 5c) and reflected in the clinical outcome in terms of PFS (Fig. 5d; high openness versus closed openness; median PFS 7.6 months versus 1.6 months; $P < 0.001$). Taken together, openness of chromatin at specific genomic positions of baseline circulating CD8$^+$ T cells predicted better clinical outcome to pembrolizumab in terms of PFS in both discovery cohort and validation cohort.

**Discussion**
In this study, we demonstrated that openness of chromatin in circulating CD8$^+$ T cells predicts better treatment outcome to pembrolizumab in mGC patients. All EBV-positive and MSI-H patients who responded to pembrolizumab had considerable openness in their circulating CD8$^+$ T cell's chromatin. Currently, there is no perfect biomarker which predicts response to ICIs. One of the working hypotheses is the influence of circulating CD8$^+$ T cells which need to be actively recruited to tumor sites upon ICI treatment but also functionally activated. These circulating CD8$^+$ T cells may be epigenetically imprinting their traces under various chronic stresses during cancer development or progression[7].

In this study, we analyzed circulating CD8$^+$ T cells in known responders to pembrolizumab using samples from phase II trial[5]. Then, we tested the established analysis algorithm in 52 GC patients who received pembrolizumab as practice. EBV positivity and MSI-H are two pathologic factors to predict response to immune check point inhibitors in GC. However, this subset is relatively rare (less than 5% of all metastatic GC patients) but the response to ICI is also observed in non-EBV, MSS subset. In addition, even within EBV-positive or MSI-H patient cohort, the response rate is 50% which suggests that there are other influential factors to determine response to immune checkpoint inhibitors. Of 13 non-EBV, MSS responders, 11 (86%) patients had high chromatin openness in their circulating CD8$^+$ T cells. Although definitive conclusion cannot be drawn from this study due to small sample size, this is the first study to demonstrate that circulating T-cell characteristics may contribute in predicting response to ICI.

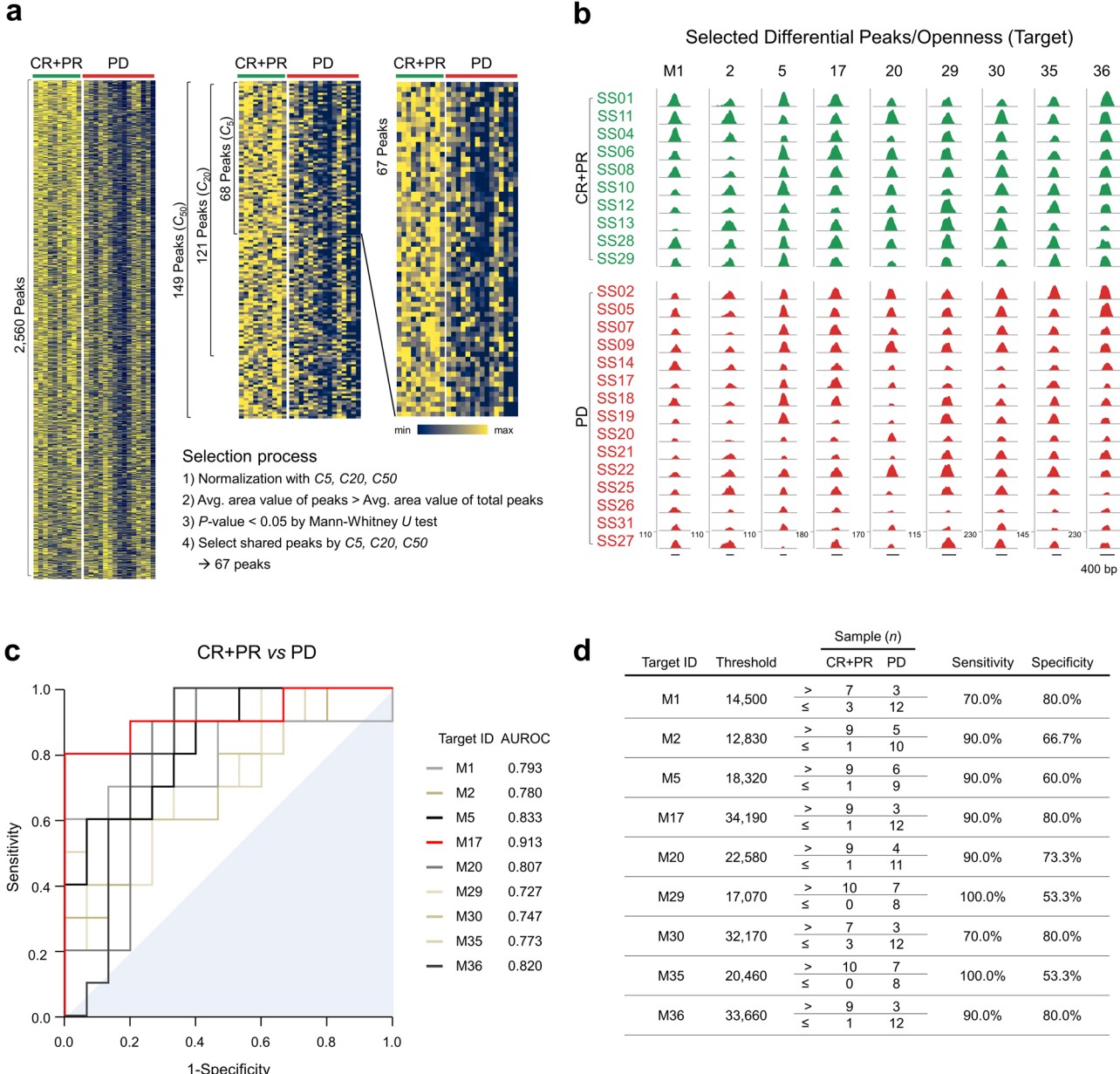

**Fig. 3 Selection of control and differential peaks, followed by normalization. a** We selected 2,560 differential peaks that distinguished between a responder group (complete response plus partial response, CR + PR) and a non-responder group (progressive disease, PD) using three peak callers: the HOMER suite, MACS2, and CisGenome. These peaks were normalized by control peaks (Supplementary data 1), then subjected to selection procedures. Chromatin accessibility heatmaps with normalized peak area values by 20 controls (rows) and patients (columns) were displayed as a representative. *C5*, *C20* and *C50* indicate 5 controls, 20 controls and 50 controls, respectively. Peaks were ranked by relative distance between the two groups as described in Methods. Relative minimum and maximum area values are indicated in navy and yellow, respectively. **b** Genome browser tracks show nine differential peaks (i.e., openness) as targets in a responder group (CR + PR, green) and a non-responder group (PD, red). Numbers above each peak denote target IDs. The *y*-axis shows the adjusted read count, calculated using a normalization factor (*F*). Scale bar indicates 400 base pairs (bp) of nucleotides. **c** Receiver operating characteristic (ROC) curves for the nine targets in the discovery cohort. Adjacent are target IDs with the areas under the ROCs (AUROCs) of the nine targets. The area under the diagonal reference line is filled. **d** Determination of threshold values for the nine targets. Target IDs are arranged in descending order of relative mean difference between responder (CR + PR) and non-responder (PD) groups. For each target, the threshold value indicates the optimized guideline calculated by the online Cutoff Finder tool, allowing prediction of response to anti-PD-1 therapy. Responders and non-responders were counted as the number of samples above or below the threshold, respectively, to determine sensitivity and specificity.

It has been shown in the previous report[17] that chromatin states defined tumor-specific T-cell dysfunction and reprogramming. They have shown that using ATAC-seq and RNA-seq of TILs and CD8+ T cells, cancer patients who do not respond to ICIs had lymphocytes in a plastic state which are not amenable to reprogramming. In our study, when openness of chromatin at specific genomic position is high in circulating CD8+ T cells, patients responded to pembrolizumab. We speculate that these patients have CD8+ T cells in a plastic state with less dysfunction and amenable to reprogramming upon blockade of PD-1. In addition, the enrichment of the transcription factor binding motifs located within significant nine representative peaks

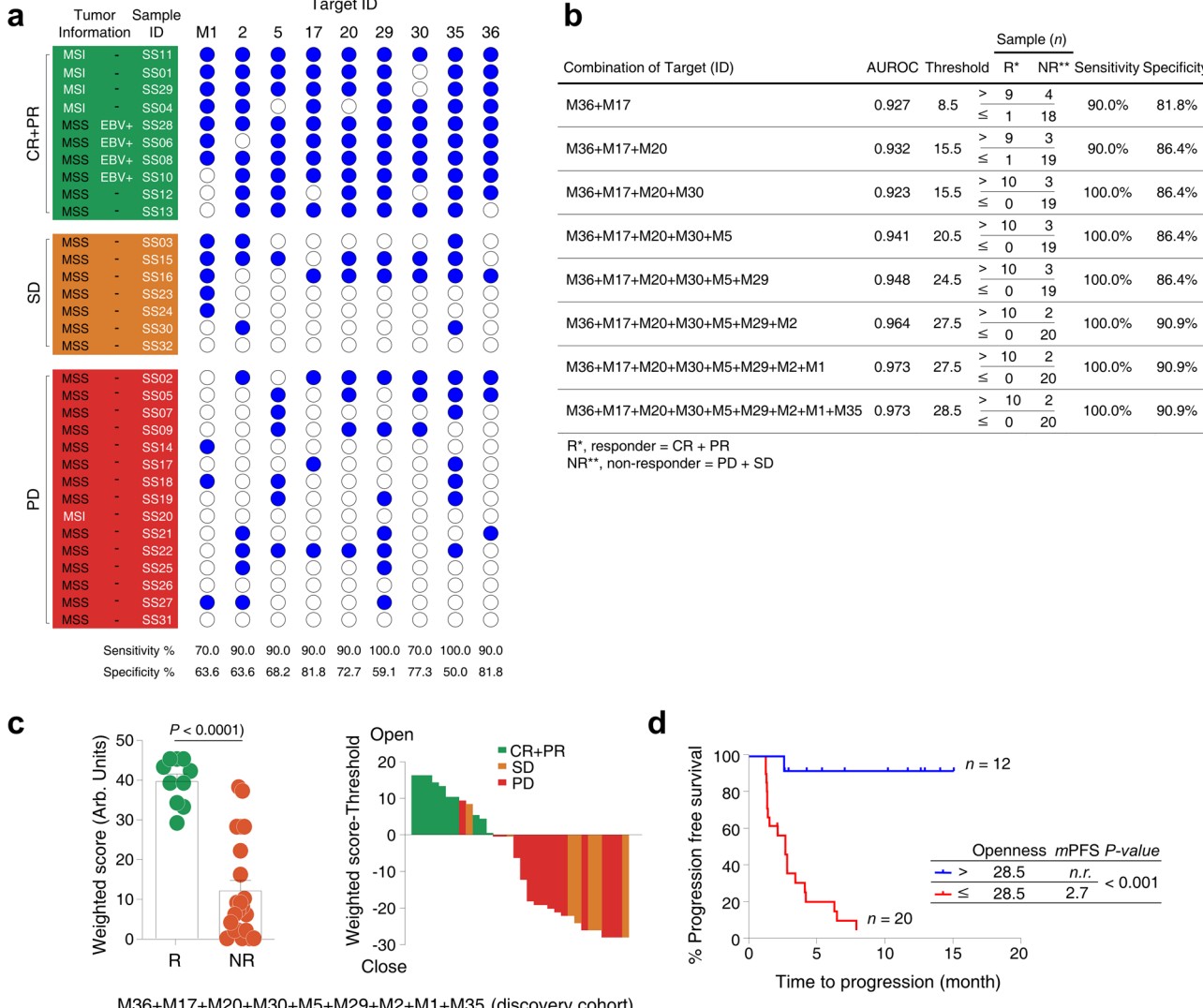

**Fig. 4 Correlation between chromatin openness of circulating CD8⁺ T cells and treatment response to anti-PD-1 therapy (discovery cohort, *n* = 32).** **a** Patients with normalized area values above and below the threshold are indicated by filled (blue) and open circles, respectively, at each target peak. Sensitivity and specificity were calculated for a responder group (R, CR + PR) and a non-responder group (NR, PD + SD). MSI: microsatellite instability; MSS: microsatellite stability; EBV: Epstein–Barr virus. PD: progression disease; SD: stable disease; PR: partial response; CR: complete response. **b** Nine targets in descending order of accuracy (ACC) were statistically evaluated in combination. After normalized areas above and below the threshold had been converted to 1 and 0, respectively, these binary values were multiplied by values ranging from 1 (lowest) to 9 (highest) on the accuracy rank, to obtain a weighted score. Instead of normalized area values, the weighted scores were used for further ROC analysis and to calculate a threshold for each combination of the nine targets using Cutoff Finder. AUROC indicates the area under receiver operating characteristic. **c** Openness scores (weighted score) for a responder group (R, CR + PR) and a non-responder group (NR, PD + SD) with normalized area values were displayed in a dot plot (left graph). Bars and error bars indicate the mean area value and SEM for each group (*n* = 10 from R and *n* = 22 from NR). *P*-value < 0.0001 was obtained using the two-sided Mann–Whitney U test. Arb. units indicate arbitrary units. Waterfall plots according to the openness of circulating CD8⁺ T cells were displayed in the middle panel. Green: CR + PR; Orange: SD; Red: PD. **d** Threshold value of combination with the nine targets was used to determine clinical outcomes of anti-PD-1 therapy using progression-free survival (PFS) curve. Tumor progression scores for patients above (*n* = 12) and below (*n* = 20) the threshold are shown in blue and red, respectively. Median PFS (*m*PFS) time was calculated using Kaplan–Meier survival analysis. Survival curves of the two groups were compared using the two-sided log-rank (Mantel–Cox) test in a single comparison (*P* < 0.0001). *n.r.*, not reached.

(Supplementary Fig. 8) could be linked to activation of T cell receptor signaling pathway such as nuclear factor of activated T cell (NFAT), adaptor-related protein complex 1 (AP-1) and nuclear factor κB (NF-κB). These findings suggest that chromatin openness of specific genomic positions in exhausted CD8⁺ T cells is probably associated with the restoration of T-cell functions by PD-1 blockade. Thus, we plan to further characterize CD8⁺ T cells with single cell RNA-seq and ATAC-seq analysis to identify gene-specific targets for activation of CD8⁺ T cells upon ICI treatment.

Taken together, the chromatin state of circulating CD8⁺ T cells was influential on response to ICI in mGC patients. Tremendous effort has been focused on profiling patient's tumor; nevertheless, we demonstrated that mGC patients with high openness at specific genomic positions of their circulating CD8⁺ T-cell chromatin responds better to pembrolizumab with longer PFS. The functional activation of the CD8⁺ T cells with high open chromatin at specific genomic positions is currently being investigated. While further validation of the findings is underway,

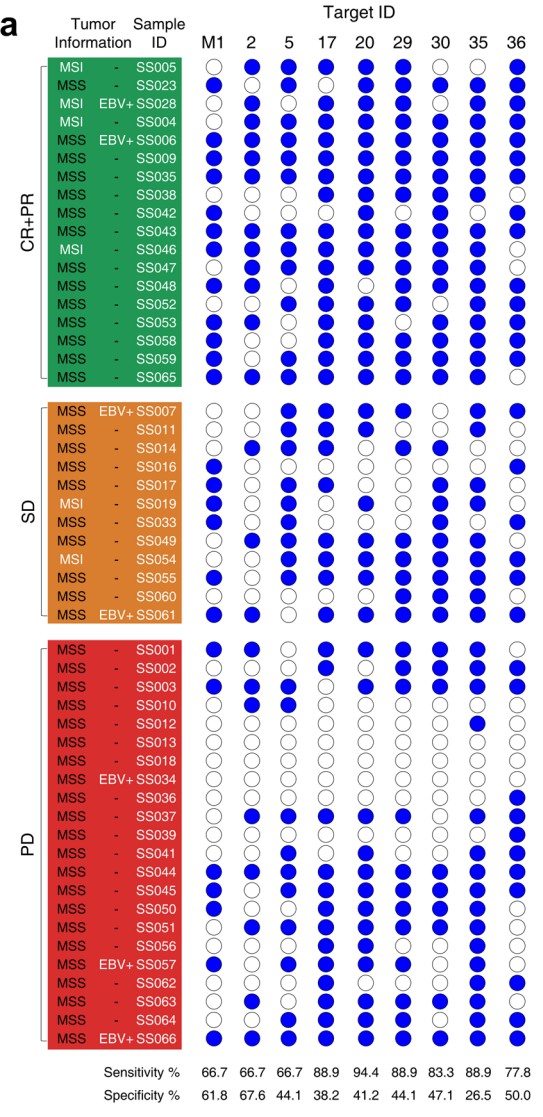

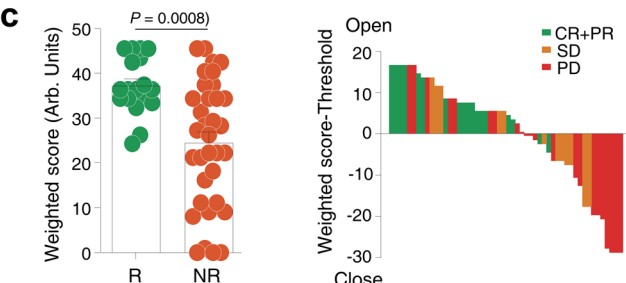

**b**

| Combination of Target (ID) | AUROC | Threshold | Sample (n) | | | Sensitivity | Specificity |
|---|---|---|---|---|---|---|---|
| | | | | R* | NR** | | |
| M36+M17 | 0.626 | 8.5 | > | 14 | 17 | 77.8% | 50.0% |
| | | | ≤ | 4 | 17 | | |
| M36+M17+M20 | 0.688 | 15.5 | > | 14 | 13 | 77.8% | 61.8% |
| | | | ≤ | 4 | 21 | | |
| M36+M17+M20+M30 | 0.730 | 15.5 | > | 18 | 18 | 100.0% | 47.1% |
| | | | ≤ | 0 | 16 | | |
| M36+M17+M20+M30+M5 | 0.720 | 20.5 | > | 18 | 17 | 100.0% | 50.0% |
| | | | ≤ | 0 | 17 | | |
| M36+M17+M20+M30+M5+M29 | 0.700 | 24.5 | > | 17 | 16 | 94.4% | 52.9% |
| | | | ≤ | 1 | 18 | | |
| M36+M17+M20+M30+M5+M29+M2 | 0.706 | 27.5 | > | 16 | 14 | 88.9% | 58.8% |
| | | | ≤ | 2 | 20 | | |
| M36+M17+M20+M30+M5+M29+M2+M1 | 0.718 | 27.5 | > | 16 | 14 | 88.9% | 58.8% |
| | | | ≤ | 2 | 20 | | |
| M36+M17+M20+M30+M5+M29+M2+M1+M35 | 0.717 | 28.5 | > | 16 | 14 | 88.9% | 58.8% |
| | | | ≤ | 2 | 20 | | |

R*, responder = CR + PR
NR**, non-responder = PD + SD

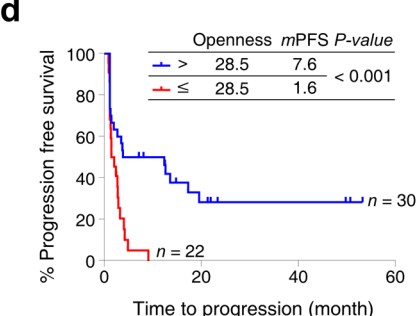

**Fig. 5 Correlation between chromatin openness of circulating CD8$^+$ T cells and treatment response to anti-PD-1 therapy (validation cohort, n = 52). a** Patients with normalized area values above and below the threshold are indicated by filled (blue) and open circles, respectively, at each target peak. Sensitivity and specificity were calculated for a responder group (R, CR + PR) and a non-responder group (NR, PD + SD). MSI: microsatellite instability; MSS: microsatellite stability; EBV: Epstein–Barr virus. PD: progression disease; SD: stable disease; PR: partial response; CR: complete response. **b** Nine targets in descending order of accuracy (ACC) were statistically evaluated in combination. After normalized areas above and below the threshold had been converted to 1 and 0, respectively, these binary values were multiplied by values ranging from 1 (lowest) to 9 (highest) on the accuracy rank, to obtain a weighted score. Instead of normalized area values, the weighted scores were used for further ROC analysis and to calculate a threshold for each combination of the nine targets using Cutoff Finder. AUROC indicates the area under receiver operating characteristic. **c** Openness scores (weighted score) for a responder group (R, CR + PR) and a non-responder group (NR, PD + SD) with normalized area values were displayed in a overlaid dot plot and bar graph (left graph). Bars and error bars indicate the mean area value and SEM for each group (n = 18 from R and n = 34 from NR). P-value = 0.0008 was obtained using the two-sided Mann–Whitney U test. Arb. units indicate arbitrary units. Waterfall plots according to the openness of circulating CD8$^+$ T cells were displayed in the middle panel. Green: CR + PR; Orange: SD: Red: PD. **d** Threshold value of combination with the nine targets was used to determine clinical outcomes of anti-PD-1 therapy using progression-free survival (PFS) curve. Tumor progression scores for patients above (n = 30) and below (n = 22) the threshold are shown in blue and red, respectively. Median PFS (mPFS) time was calculated using Kaplan–Meier survival analysis. Survival curves of the two groups were compared using the log-rank (Mantel–Cox) test in a single comparison (P = 0.0005). n.r., not reached.

epigenetic landscape of CD8$^+$ T cells should be actively investigated in ICI-treated cancer patients.

## Methods

**Study design and participants**. This study was approved by the Institutional Review Boards of Seoul National University Hospital (no. 0905-014-280) and Samsung Medical Center (no. 2015-09-053). Written informed consent was obtained from all patients, in accordance with the Declaration of Helsinki. Patients participating in the clinical trial (clinicaltrials.gov identifier NCT#02589496)[5] were designated as the discovery cohort. For validation cohort, we prospectively recruited mGC patients (n = 52) who were undergoing anti-PD-1 therapy with pembrolizumab and collected peripheral blood before treatment. Clinical characteristics of all patients with mGC are provided in Supplementary Tables 1 and 2.

Tumor specimen microsatellite instability-high (MSI-H) or EBV positivity status information was obtained from our previous report[5]. Written informed consent was obtained from all patients, in accordance with the Declaration of Helsinki.

**Flow cytometry and cell sorting**. Peripheral blood mononuclear cells (PBMCs) were purified from heparinized peripheral blood using a Ficoll–Histopaque gradient (1.077 g/mL; GE Healthcare Life Sciences, Piscataway, NJ, USA) and stored in liquid nitrogen after suspension in freezing media including 50% fetal bovine serum, 10% dimethyl sulfoxide, and 40% RPMI-1640 (Invitrogen, Carlsbad, CA, USA) until further analysis[30]. To obtain CD8[+] T cells, thawed PBMCs were stained with appropriate Abs and sorted using the FACSAria III cell sorter (BD Biosciences, San Jose, CA, USA).

Abs used for flow cytometry and sorting included Alexa 700–anti-CD8 (1:100; Cat.No. 557945; clone, RPA-T8), allophycocyanin–anti-PD-1 (1:5; Cat. No. 558694; clone, MIH4) and FITC–anti-Ki-67 (1:25; Cat. No. 51-36524X; clone, B56) (all from BD Biosciences). Staining intracellular molecules was performed after permeabilization using a Cytofix/Cytoperm kit (BD Biosciences). Stained cells were analyzed using an LSR II flow cytometer (BD Biosciences) with the FACSDiva software. Data were analyzed using the FlowJo software (TreeStar, Ashland, OR, USA).

**ATAC-seq library preparation and high-throughput sequencing**. ATAC-seq was performed as described previously[8]. In details, viable sorted 50,000 CD8[+] T cells (Supplementary Fig. 1b) were washed with cold 1x PBS at 4°C and harvested via centrifugation at 500 × g for 5 min at 4°C. These cells were resuspended in 50 μL of cold lysis buffer (10 mM Tris-HCl, pH 7.4, 10 mM NaCl, 3 mM MgCl2, 0.1% IGEPAL CA-630) and nuclei were isolated by centrifugation at 500 × g for 10 min at 4°C, followed by the transposition reaction in the transposase reaction mix (25 μL 2× TD buffer, 2.5 μL Tn5 transposase (Illumina) and 22.5 μL nuclease-free water) and incubated at 37°C for 30 min with mild agitation. DNA from the transposition reaction were purified via a Qiagen PCR cleanup kit. PCR amplification was performed using indexing primers (Supplementary Table 9) from the Nextera kit (Illumina, San Diego, CA, USA). The optimal number of cycles in each sample was determined via qPCR to stop the amplification before saturation of indexed DNA libraries. After preparation, each library was quantified using a Bioanalyzer (Agilent Technologies, Santa Clara, CA, USA), based on the proportion of index primer contamination interfering with the Illumina sequencer (Illumina NextSeq500). Each library was then verified by quantitative PCR using the KAPA Library Quantification Kit (Roche Applied Science, Basel, Switzerland). The pool of indexed libraries was sequenced using Illumina NextSeq500 for 75 single-read bases and analyzed using the CASAVA (v. 1.8.2) base-calling software (Illumina). Resultant reads were checked using the FASTQC software to assess the quality of single-end or paired-end read sequences. For analysis, sequencing data of each sample required a Phred quality score and total number of reads equal to or greater than 30 (Q30) and 20 million, respectively. Detailed further analytical method for ATAC-seq is provided as below.

**Peak area calculation**. Identification of accessible chromatin regions marked as peaks was performed using the HOMER suite with the options for variable- or fixed-width peaks. As previously described[22], fixed-width peaks may be beneficial in high-quality analysis for less biased large peaks and large datasets. Enrichments beyond fixed-width peaks were excluded. During comparison of overall chromatic accessibility (openness), the loss of tags within fixed ranges can dilute differences between samples; to overcome this problem, all sample data were concatenated into a single set and peak calling was performed using the HOMER suite for variable-width peaks. Peak regions thus obtained correspond to each peak in each sample. Instead of tag enrichments, peak area, representing the sum of tag enrichment, was calculated using each bedGraph in the sample. Briefly, bedGraphs were generated using the HOMER suite with the 'makeUCSCfile' command and the '-fragLength 147 -tbp 3 -style chipseq –raw' options. The area of each peak was then calculated by summing height per base pair in the peak range using a genome browser track file. For each sample, the area of the $q$th peak region was computed as follows:

$$A_q = \sum_{p=X_q}^{Y_q} d_{pq}$$

where $d_{pq}$ is the height per base pair at position $p$ of the coordinate from start ($X_q$) to end ($Y_q$) of the $q$th peak. Finally, the peak areas were used to quantitatively compare chromatin openness.

**Control peak selection**. To determine universal controls for normalization from CD8[+] T cells for ATAC-seq by the selection of peaks with low variability across samples, ATAC-seq data including naive, effector memory, and central memory CD8[+] T cells from PBMCs (GSE89308) from healthy volunteers were used to screen control peaks for normalization[17]. All sample data were concatenated into a single set and peak calling was performed using the HOMER suite with variable-width peaks to generate peak regions. Next, the area of selected peaks from each CD8[+] T-cell subset was calculated using bedGraphs, as described above. After comparisons of naïve, effector memory, and central memory T-cell subsets, a total

of 32,485 overlapped peaks was obtained among all CD8[+] T-cell subsets using 'findOverlaps'. From these peaks, 4,798 peaks were selected that matched 100% with DNase I hypersensitive (DH) uniform peaks and were located on the TSS or 5' untranslated region (UTR)[24]. Among the selected peaks, 533 were consistent with corresponding regions of ATAC-seq from peripheral blood (PB) CD8[+] T cells used in this study. The final overlapped peaks between CD8[+] T-cell subsets (GSE89308) of healthy volunteers and CD8[+] T cells of patients with mGC (this study) satisfied the following criteria. First, peak area indicating chromatin openness was quantified by summing height per base pair within the peak range; CV was calculated using each sample peak area. In the 232 resulting ATAC peaks, CV was <0.3, demonstrating low variance; peak width was <500 bp, demonstrating uniqueness. Among these peaks, the 20 peaks with the highest average peak area were highly evolutionarily conserved among 17 species according to the CisGenome browser[31].

**Calculation of normalization factor ($F$)**. To ensure a normally distributed approximation, a discovery cohort $C_1$ was randomly selected with a sample size of $n_1 = 32$, and a validation cohort $C_2$ of size $n_2 = 52$.

For each sample, the mean height $h$ of the control peaks was computed as follows:

$$h = \frac{\sum_{i=1}^{m} k_i}{m}$$

where $m = 20$ is the number of selected control peaks in the sample and $k_i$ is the average height of the $i$th peak in the sample, calculated as follows:

$$k_i = \frac{\text{the area } A_i \text{ of the } i\text{th peak}}{\text{the base length of the } i\text{th peak}}$$

The peak area was computed as the sum of enrichment within the peak. Because the widths of selected peaks in each sample were considered equivalent, $h$ was used to represent the average amounts of selected control peaks for each sample.

For each integer pair $(a, b)$ such that $1 \le a \le b \le 2$, the normalization factor was defined as:

$$F_{ab} = \frac{S_a \left( = \frac{\sum_{i=1}^{n_a} h_{ai}}{n_a} \right)}{h_{bj}}$$

where $h_{ai}$ and $h_{bj}$ are the mean heights of the $i$th and $j$th samples in cohorts $C_a$ and $C_b$, respectively. The factor $F_{ab}$ compares the $j$th sample in cohort $C_b$ with those of cohort $C_a$. For example, $F_{11}$ compares the $j$th sample in discovery cohort $C_1$ with those in discovery cohort $C_1$; $F_{12}$ compares the $j$th sample in discovery cohort $C_1$ with those in validation cohort $C_2$. A complete description of the basis and derivation of the ATAC-seq normalization factor is provided in Supplementary Fig. 4. The areas of peak regions and the genome browser track of each sample were normalized by multiplying their normalization factors as follows:

$$\text{Normalized area} = F_{ab} \times A_q$$

where $F_{ab}$ is the normalization factor calculated from the $j$th sample in cohort $C_b$ with those in cohort $C_a$ and $A_q$ is the area of the $q$th peak in the $j$th sample in cohort $C_b$.

**Differential peak selection based on quantitative normalization**. To select differential peaks from ATAC-seq data of CD8[+] T cells in patients with mGC who were receiving anti-PD-1 therapy, a responder group was chosen that included patients with complete response (CR) and partial response (PR); a non-responder group was chosen that included only patients with progressive disease (PD). Within the discovery cohort, the responder and non-responder groups were classified based on the clinical decision at a certain time point after finishing multiple cycles. Although it was difficult to classify patients with stable disease (SD), these patients were excluded from the non-responder group and their peaks were clearly distinct from those of the responder and non-responder groups[5].

To increase the accuracy of identification of differential peaks between the two groups, three peak callers were used to identify 2,560 peaks: the HOMER suite, MACS2, and CisGenome. To quantitatively determine distinctive peaks between the responder and non-responder groups, each sample peak was normalized by multiplication with a normalization factor ($F$) calculated using three sets of controls, ranked –5, –20, and –50 (Supplementary Table 5). Selected 68, 121 and 149 peaks by three sets of controls respectively showed the normalized average area value of the cohort greater than the normalized average area value of total peaks and the mean difference in area between the responder and non-responder groups statistically significant ($p < 0.05$, Mann–Whitney $U$ test). Only 67 peaks among normalized peaks by three sets of controls satisfied these selection criteria (Supplementary Fig. 6).

To define representative peaks among 67 peaks, three selection criteria were used. 3 peaks were selected with higher average in the responder group. 3 peaks were selected with lower variance among area values in the responder group. Lastly the relative distance between the two groups were used; the area value of each peak was divided by the largest area value among all samples, and then the relative distance was calculated with the average of each group. 3 peaks with highest relative distance between two groups were selected.

**Determination of cut-off for each target**. For targets determined according to normalized area values, the Cutoff Finder online tool (http://molpath.charite.de/cutoff) was used to find the best cut-offs for use as predictive markers and response guidelines in patients who were receiving anti-PD-1 therapy[32]. The threshold was determined by comparing the responder and non-responder groups to minimize the Euclidean distance between the ROC curves. The threshold of each differential peak was determined as follows: first, the predictive performance of the threshold for each differential peak was estimated according to the area under the ROC curve (AUROC)[33]. Then, the discriminative ability of the threshold for both groups was assessed by calculating their sensitivity, specificity, and accuracy (ACC)[34]. Finally, the threshold of each differential peak was used in PFS analysis to determine the clinical outcomes of anti-PD-1 therapy.

**Optimal target combination**. Using multiple markers can increase the predictive power of a therapeutic response[35]. Therefore, nine selected peaks (target IDs) were aligned in descending order of accuracy values. To convert the normalized area value of each peak into a binary value, values above and below the threshold were set to 1 and 0, respectively. Binary values were then multiplied by a value ranging from 1 (lowest) to 9 (highest) based on the accuracy rank. These values were defined as weighted scores, and applied to each sample as normalized chromatin accessibility. The weighted scores were summed on the basis of the combination of markers in each sample. Thus, the new cut-off was calculated using Cutoff Finder for each combination of markers; the ROC analysis, discriminative ability, and PFS analysis were computed between the responder and non-responder groups.

**ATAC-seq alignment and peak calling**. FASTQ data were processed using the Trimmomatic software[36] to increase read quality prior to alignment (per-base sequence quality > 30) and then aligned to the primary assembly of the GRCh37/hg19 human reference genome (chr1~22 and chrX) using the Bowtie2 software with the parameters '-k 4 -N 1 -R 5 -end-to-end'[37]. Finally, aligned reads were converted into browser extensible data (BED) format using the 'bedtools bamtobed' parameter, and offset by +4 bp for the positive strand and –5 bp for the negative strand; Tn5 transposase occupies 9 bp during transposition[8]. Three peak callers were used to identify genome regions enriched with aligned reads: the Hyper-geometric Optimization of Motif EnRichment (HOMER) suite[25], Model-based Analysis of ChIP-Seq 2 (MACS2)[26], and CisGenome[27]. The aligned reads were used for peak calling using the HOMER suite using 'findPeaks' and the '-style factor -tbp 3 -region' option, MACS2 with 'callpeaks' and the '--nolambda --nomodel --shift 100 --extsize 200' option, and CisGenome with the 'Read extension length = 150 bp', 'Bin size = 50 bp', 'Max gap = 50 bp', and 'Min peak length = 100 bp' options. During peak calling, artificially high regions were designated blacklisted regions according to the Encyclopedia of DNA Elements (ENCODE) project. Ultra-high signal artefact regions were retrieved from the comprehensive collection (http://mitra.stanford.edu/kundaje/akundaje/release/blacklists/) and excluded from the resultant peaks. Signal tracks were generated using the HOMER suite with the 'makeUCSCfile' command and the '-fragLength 147 -tbp 3-style chipseq -raw' options, and visualized in the UCSC genome browser (https://genome.ucsc.edu/).

**ATAC-Seq quality control: transcription start site (TSS) enrichment and fraction of reads in peaks (FRiP)**. FRiP for each sample was the number of aligned reads that overlapped the peak using "findOverlaps" divided by the total number of aligned reads in the sample. To compute TSS enrichment score, each TSS coordinates from HOMER suite "hg19.tss" was extended 2000 bp in both direction and counted strand-corrected reads using "findOverlaps". The number of reads in each single base position bin was summed and divided by the average number of reads at the TSS boundary (±1900–2000 bp from TSS). Finally, the TSS enrichment score was the maximum enrichment value within ± 50 bp of TSS after smoothing with a moving average for a window of 51 bps.

**Statistical analyses**. Data were expressed as means ± standard errors of the mean (SEMs) and compared using the Mann–Whitney $U$ test, Wilcoxon signed rank test, and one-way ANOVA test. Progression-free survival (PFS) and median survival were determined using the Kaplan–Meier method. Groups were compared using the log-rank (Mantel–Cox) test. All statistical analyses were performed using GraphPad Prism 6.01 (GraphPad Software, La Jolla, CA, USA) and R statistical softwares (R Foundation for Statistical Computing, Vienna, Austria, http://www.R-project.org/). The Cutoff Finder (http://molpath.charite.de/cutoff/) was used to determine an optimal threshold for chromatin openness to classify responder and non-responder patients who were receiving anti-PD-1 therapy. ROC analysis measured the predictive accuracy of selected peaks. In all analyses, $P < 0.05$ was considered to indicate statistical significance.

**Reporting summary**. Further information on research design is available in the Nature Research Reporting Summary linked to this article.

## Data availability

The ATAC-seq data used in this study are available in the Gene Expression Omnibus (GEO) database under accession code GSE142604. The public data used in this study are available in the GEO database under accession code GSE89308, GSE74912, GSE118204, and GSE117685. The remaining data supporting the findings of this study are available within the Article, Supplementary Information or available from the corresponding authors upon request.

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

## Acknowledgements

This work was supported in part by the Creative-Pioneering Researchers Program through Seoul National University (to H.R.K.); the Korea Healthcare Technology Research and Development Project, funded by the Ministry for Health, Welfare, and Family Affairs of Korea (HI17C0534 to H.R.K.); the Small Grant Exploratory Research (SGER) through the National Research Foundation of Korea (NRF) Grant funded by the Ministry of Education (2017R1D1A1A02019177 to H.M.S.). We thank J. Park at Core Research Facilities of Seoul National University College of Medicine for help with sorting human CD8+ T-cell subsets, Prof. A.K. Park at Sunchon National University and M. Choi at Seoul National University, Republic of Korea for valuable discussions and insightful comments. The original manuscript was greatly improved by their critical comments.

## Author contributions

H.M.S., J.L and H.-R.K. conceived and designed the project. G.K. and H.M.S. performed the ATAC–seq library construction and sequencing. G.K., H.M.S., M.K., S.K., J.C., S.Y., D.L., S.W.C. and S.T.K. analyzed the data. G.K., J.H.S., M.J.K. and H.-R.K. collected the human PBMCs, and performed CD8+ T-cell separation and flow cytometric analysis. J.L, H.M.S. and H.-R.K. supervised the project or various experiments. G.K., S.K., J.H.S., J.L, H.M.S. and H.-R.K. wrote the manuscript with the help from all authors. H.-R.K. had full access to all data in the study and took responsibility for the integrity of the data, as well as for the manuscript.

## Competing interests

The authors declare no competing interests.
