## [Peer Review File · Nature Communications]

Editorial note: Parts of this Peer Review File have been redacted to maintain patient confidentiality. Please refer to Supplementary Tables 1-2 in the Supplementary Information file for patients characteristics”.

REVIEWER COMMENTS

Reviewer #1 (Remarks to the Author):

This paper focused on host-tumor interactions that can alter epigenetic characteristics of peripheral blood CD8+ T-cells that can predict for response to immune checkpoint inhibitors (ICIs). Specifically, using a discovery cohort from a clinical trial investigating pembrolizumab in refractory advanced gastric cancer (GC) patients and a validation cohort of prospectively recruited cohort of mGC patients treated with pembrolizumab, PBMCs were isolated and baseline chromatin openness in CD8+ T-cells as determined by ATAC-seq predicted for response and PFS to anti-PD-1 therapy. This is an interesting study exploring a novel concept that epigenetic changes, in this instance, the openness of chromatin at specific genomic positions in circulating CD8+ T-cells can affect response to PD-1 blockade, can putatively be associated to increased immune cell plasticity and susceptibility to reprogramming with PD-1 blockade.

The strengths of this study appear to be the robust pipeline of discovery and validation of chromatin openness as assessed by ATAC-seq with publicly available datasets for future investigators to attempt to emulate, particularly if this novel predictive biomarker is to be implemented in large, prospective study designs of ICIs.

The authors do mention that all EBV-positive and MSI-H patients had high chromatin openness. Can the authors comment on the frequency of EBV-negative and MSS patients with high chromatin openness and to what degree of response or PFS benefit do these patients compare to low openness and EBV-positive/MSI-H patients? Additionally, what was the relationship with PD-L1 status and chromatin openness? Can the authors comment if chromatin openness is affected by prior treatment with chemotherapy, as this was a heavily pretreated population of patients?

Reviewer #2 (Remarks to the Author):

Shin et al analyzed chromatin accessibility in peripheral blood CD8 T cells in 2 cohorts (32 and 22 patients) of gastric cancer patients treated with PD-1 blockade. Chromatin accessibility of 9 regions predicted response to treatment with 100% sensitivity and 90% specificity. These results are novel since ATAC-seq has not been extensively performed in peripheral blood of checkpoint patients, and this is the first demonstration that pre-therapy T cell epigenetics can inform response and resistance. However, I have several concerns that temper my enthusiasm. First is the lack of any discussion regarding whether this is purely a biomarker study, or whether there is some biological meaning to these findings and peaks. If this is purely a biomarker study, there should be a more rigorous discussion of sensitivity/ specificity in the validation cohort and how the authors imagine this being used clinically to select patients for therapy. Second, the authors should also discuss the patient heterogeneity present in this study (almost all responders are MSI/EBV) and whether this limits the application of this assay to responders without those clinical features. Third, there is the potential for overfitting, since there are thousands of ATAC-seq peaks, and therefore the authors should transparently discuss why they focused only on the 9 differential peaks (rather than all 67).

Major comments:

1. The authors should describe in detail their experimental ATAC-seq protocol, and quality control measurements for each sample. QC statistics of interest are TSS scores and fraction reads in peaks metrics for each sample. Sample quality filtering is not described at all in the results section- did all samples pass TSS/FRIP score filtering? How many cells were assayed in each sample?

2. The step 2 filtering process in Fig 3A is quite difficult to understand. The authors should

carefully describe what exactly is done to go from 121 to 67 peaks, and give examples of features of the peaks that are filtered out at this stage.

3. A major issue is the progression from 67 peaks discovered in the unbiased manner described in Fig 3A to 9 peaks described in Figures 4 and 5. Why were 9 peaks chosen and not 10 or 11? What happens to the performance (particularly specificity) of the test if all 67 peaks are included? How do all 67 peaks perform in the validation cohort? The authors should provide the sensitivity/specificity data for each of the 67 peaks that were identified.

4. The authors should describe the identity of the 67 peaks in more detail. I might suggest a GREAT analysis (great.stanford.edu) of nearby genes, enriched pathways, etc. Further, are these peaks memory or naïve T cell specific (or common peaks)? Are they open in other cell types? The authors could compare to some of the reference datasets that they describe earlier in the manuscript.

5. The discussion regarding plastic dysfunctional cells is provocative, but has no support from the presented data. The fact that AP-1 or NFAT motifs are present in the differential peaks may represent some relation to T cell dysfunction, or more likely, just the fact that they are indeed T cells where these TFs are common and active. In other words, AP-1 and NFAT are active in normal T cell development and activation, and in exhaustion, and there is no way to know if the presence of binding sites in a peak represents an activation or exhaustion signature. Are the 67 peaks enriched in dysfunction or exhaustion-associated genes or TFs?

In particular, the authors make a broad statement on line 323 that 'These findings suggest that chromatin openness of specific genomic positions in exhausted CD8+ T cells is probably associated with the restoration of T-cell functions by PD-1 blockade.' This statement is not presently supported by any of the data in the manuscript and should be removed.

6. Finally, there is a concern that the data is confounded by EBV and MSI status. For example, if I examine responders who are EBV/MSI negative, the sensitivity and specificity appear to go down. I.e Fig 5C, only 2/5 high openness patients that are EBV/MSI- respond to therapy. Is the present peak set and chromatin analysis applicable to all GC patients, or only those that are EBV/MSI +? The authors should run an analysis on EBV/MSI- patients and report sensitivity and specificity in both cohorts.

Reviewer #3 (Remarks to the Author):

In this manuscript, the relationship between epigenetic characteristics of baseline CD8+ T cells and anti-PD1 treatment is considered. A clinical trial (NCT#02589496) of 32 mGC patients is considered as the discovery cohort and another clinical trial of 22 patients who received pembrolizumab is considered as the validation cohort. However, there are no any biographical data of patients except patients' age. I have the following questions:

1. Are the characteristics of baseline CD8+ T cells relative to patient biographical variables?
2. As a biomarker, the cut-of-points in the validation cohort should be the same as that in the discovery cohort. However, the thresholds in the discovery cohort (Figure 4B) are different with the thresholds in the validation cohort (Figure 5B). It is also in Figure 4D (the cut-of-point =28.5) and in Figure 5D (the cut-of-point = 26.5).
3. The sample size in the validation cohort is not enough.

Responses to Reviewers' Comments

(Authors' Response/Action) We appreciate your comments very much. We revised the manuscript in accordance with your advice. In the following, we addressed the reviewers' comments point-by-point and made other necessary corrections. The revision made to this manuscript are marked in yellow-highlights.

REVIEWER COMMENTS

Reviewer #1 (Remarks to the Author):

This paper focused on host-tumor interactions that can alter epigenetic characteristics of peripheral blood CD8⁺ T-cells that can predict for response to immune checkpoint inhibitors (ICIs). Specifically, using a discovery cohort from a clinical trial investigating pembrolizumab in refractory advanced gastric cancer (GC) patients and a validation cohort of prospectively recruited cohort of mGC patients treated with pembrolizumab, PBMCs were isolated and baseline chromatin openness in CD8⁺ T-cells as determined by ATAC-seq predicted for response and PFS to anti-PD-1 therapy. This is an interesting study exploring a novel concept that epigenetic changes, in this instance, the openness of chromatin at specific genomic positions in circulating CD8⁺ T-cells can affect response to PD-1 blockade, can putatively be associated to increased immune cell plasticity and susceptibility to reprogramming with PD-1 blockade.

The strengths of this study appear to be the robust pipeline of discovery and validation of chromatin openness as assessed by ATAC-seq with publicly available datasets for future investigators to attempt to emulate, particularly if this novel predictive biomarker is to be implemented in large, prospective study designs of ICIs.

The authors do mention that all EBV-positive and MSI-H patients had high chromatin openness. Can the authors comment on the frequency of EBV-negative and MSS patients with high chromatin openness and to what degree of response or PFS benefit do these patients compare to low openness and EBV-positive/MSI-H patients?

(Authors' Response/Action) We are grateful for your kind comments. In discovery cohort, although most EBV-positive patients and MSI-H patients who responded to pembrolizumab had high openness of chromatin at specific genomic positions of baseline CD8⁺ T cells (Fig. 4A), one MSI-H patient (SS #20) with resistance to pembrolizumab had closed chromatin at the corresponding genomic positions of baseline CD8⁺ T cells (Fig. 4A). MSI-H and EBV-positive are good marker for prediction of anti-PD-1 therapy, but we still have these exceptions in the new validation cohort.

To address reviewer's comment of regarding the relationship between EBV status and MSI-H and survival benefit in the response to anti-PD-1 therapy in gastric cancer patients, total 84 patients in the discovery and the validation cohorts were pooled. This entire patient pool without considering any clinical features was used to perform a log-rank test with a threshold by the chromatin openness to analyze the progression-free survival (PFS). This analysis can determine whether this guideline by chromatin openness provide a predictive benefit of patient survival in the response to the anti-PD-1 therapy, for the combination of selected markers. PFS for mGC patients with high chromatin openness of circulating CD8⁺ T cells was > 11.7 months while PFS for low openness was 2.1~2.6 months ($P < 0.001$) as follows. mGC patients with high openness chromatin of CD8⁺ T cells demonstrated substantially longer PFS following anti-PD1 treatment compared to those with closed chromatin.

Chromatin Openness		Discovery + Validation (n=84)				Log-rank test P -value
		Open (Area > Threshold)		Close (Area < Threshold)		
		m PFS	n	m PFS	n	
	M36	11.7	44	2.6	40	0.0010
Combinations	M36+M17	11.7	44	2.6	40	0.0010
	M36+M17+M20	12.8	39	2.6	45	< 0.0001
	M36+M17+M20+M30	11.7	49	2.6	35	< 0.0001
	M36+M17+M20+M30+M5	11.9	48	2.3	36	< 0.0001
	M36+M17+M20+M30+M5+M29	11.9	46	2.3	38	< 0.0001
	M36+M17+M20+M30+M5+M29+M2	12.8	42	2.1	42	< 0.0001
	M36+M17+M20+M30+M5+M29+M2+M1	12.8	42	2.1	42	< 0.0001
	M36+M17+M20+M30+M5+M29+M2+M1+M35	12.8	42	2.1	42	< 0.0001

Next, we considered how EBV-positive or MSI-H in the tumor tissue change the prediction of patient's survival. Patients selected were divided into two groups: one groups with MSS and EBV-negative and the other group with EBV-positive or MSI-H from two cohorts. mGC patients with EBV-positive or MSI-H demonstrated significantly longer PFS following anti-PD-1 therapy (median prolonged PFS not reached) when compared to those with closed chromatin as shown in the below table. This indicates that even in patients with open chromatin at specific genomic position of CD8⁺ T cells, EBV-positive or MSI-H significantly contribute to the patient's survival.

Tumor Info. Chromatin Openness		Discovery + Validation (n=84)				Log-rank test
		MSS and EBV-		MSI or EBV+		
		Open (Area > Threshold)		Open (Area > Threshold)		P-value
		mPFS	n	mPFS	n	
Combinations	M36	2.7	28	u.d.	16	0.0008
	M36+M17	2.7	28	u.d.	16	0.0008
	M36+M17+M20	3.3	23	u.d.	16	0.0032
	M36+M17+M20+M30	2.6	32	u.d.	17	0.0005
	M36+M17+M20+M30+M5	2.6	31	u.d.	17	0.0007
	M36+M17+M20+M30+M5+M29	2.6	29	u.d.	17	0.0011
	M36+M17+M20+M30+M5+M29+M2	3.5	25	u.d.	17	0.0035
	M36+M17+M20+M30+M5+M29+M2+M1	3.5	25	u.d.	17	0.0035
	M36+M17+M20+M30+M5+M29+M2+M1+M35	3.5	25	u.d.	17	0.0035

The number of patients with EBV-negative or MSS are only 1.4 to 1.8 times EBV-positive or MSI of patients with high chromatin openness. Therefore, it can be considered that patients with high chromatin openness have more survival benefits for anti-PD-1 therapy than patients with closed chromatin, only because of patients with EBV-positive or MSI-H. However, interestingly, patients with MSS and EBV-negative who have high chromatin openness, have the benefit for the survival compared to the group of patients' low chromatin openness (closed chromatin) at specific genomic positions. Thus, this suggests that regardless of EBV-positive or MSI-H, chromatin openness is sufficient to predict responsiveness to anti-PD-1 therapy.

Tumor Info. Chromatin Openness		Discovery + Validation (n=84)				Log-rank test
		MSS and EBV-		All		
		Open (Area > Threshold)		Close (Area < Threshold)		P-value
		mPFS	n	mPFS	n	
Combinations	M36	2.7	28	2.6	40	0.2221
	M36+M17	2.7	28	2.6	40	0.2221
	M36+M17+M20	3.3	23	2.6	45	0.0687
	M36+M17+M20+M30	2.6	32	2.6	35	0.0510
	M36+M17+M20+M30+M5	2.6	31	2.3	36	0.0294
	M36+M17+M20+M30+M5+M29	2.6	29	2.3	38	0.0169
	M36+M17+M20+M30+M5+M29+M2	3.5	25	2.1	42	0.0015
	M36+M17+M20+M30+M5+M29+M2+M1	3.5	25	2.1	42	0.0015
	M36+M17+M20+M30+M5+M29+M2+M1+M35	3.5	25	2.1	42	0.0015

Additionally, what was the relationship with PD-L1 status and chromatin openness? Can the authors comment if chromatin openness is affected by prior treatment with chemotherapy, as this was a heavily pretreated population of patients?

(Authors' Response/Action) We appreciate the reviewer for bring this to our attention. The interaction between programmed cell death-1 (PD-1) and its ligand (PD-L1) exerts a considerable effect on immune escape, tumor progression, and metastasis (Freeman *et. al.*, J Exp Med 2000). Thus, this idea led to PD-1

blocking Abs for the purpose of killing tumor, and ultimately approved as a treatment. As the reviewer knows, the expression levels of PD-L1 in tumor tissues is unreliable to predict the anti-PD-1 treatment response, even the FDA has approved PD-L1 immunohistochemistry as a companion diagnostic indicator in certain types of cancer treatment. According to the reviewer's question, we tried to determine correlation between PD-L1 expression status in tumor tissues and the chromatin openness of CD8⁺ T cells, 77 patients with PD-L1 combined positive score (CPS) were analyzed by Fisher's exact test. At CPS ≥ 1 cut-off, 51 patients among 77 patients were positive for PD-L1. Then the significance of the association (contingency) was calculated for each biomarker. The data showed that 3 peaks (M2, M17, M29) among 9 representative peaks were related between PD-L1 in tumor and the high chromatin openness of CD8⁺ T cells. As expected, the expression of PD-L1 is not considered to be highly correlated with high chromatin openness at specific genomic position of CD8⁺ T cells, which predicts reactivity to anti-PD-1 therapy fairly well.

Tumor info	Discovery + Validation				Fisher's exact test	
	PD-L1 ⁺	PD-L1 ⁺	PD-L1 ⁻	PD-L1 ⁻	P -value	n
Chromatin Openness	Open (Area > Threshold)	Close (Area \leq Threshold)	Open (Area > Threshold)	Close (Area \leq Threshold)		
M1	30	21	10	16	0.0997	77
M2	31	20	6	20	0.0020	77
M5	29	22	13	13	0.6326	77
M17	38	13	11	15	0.0113	77
M20	36	15	13	13	0.0862	77
M29	40	11	12	14	0.0091	77
M30	31	20	13	13	0.4663	77
M35	41	10	17	9	0.1705	77
M36	32	19	10	16	0.0547	77

Reviewer #2 (Remarks to the Author)

Shin et al analyzed chromatin accessibility in peripheral blood CD8 T cells in 2 cohorts (32 and 22 patients) of gastric cancer patients treated with PD-1 blockade. Chromatin accessibility of 9 regions predicted response to treatment with 100% sensitivity and 90% specificity. These results are novel since ATAC-seq has not been extensively performed in peripheral blood of checkpoint patients, and this is the first demonstration that pre-therapy T cell epigenetics can inform response and resistance. However, I have several concerns that temper my enthusiasm. First is the lack of any discussion regarding whether this is purely a biomarker study, or whether there is some biological meaning to these findings and peaks. If this is purely a biomarker study, there should be a more rigorous discussion of sensitivity/ specificity in the validation cohort and how the authors imagine this being used clinically to select patients for therapy. Second, the authors should also discuss the patient heterogeneity present in this study (almost all responders are MSI/EBV) and whether this limits the application of this assay to responders without those clinical features. Third, there is the potential for overfitting, since there are thousands of ATAC-seq peaks, and therefore the authors should transparently discuss why they focused only on the 9 differential peaks (rather than all 67).

(Authors' Response/Action) We are grateful for your kind comments and have tried our best to revise the manuscript accordingly.

“If this is purely a biomarker study, there should be a more rigorous discussion of sensitivity/ specificity in the validation cohort and how the authors imagine this being used clinically to select patients for therapy.”

In this study, we first analyzed circulating CD8⁺ T cells in known responders to pembrolizumab using samples from phase II trial. Then, we tested the established analysis algorithm in 52 GC patients who received pembrolizumab as practice. EBV positivity and MSI-H are two strong pathologic factors to predict response to immune check point inhibitors in GC. However, this subset is relatively rare (less than 5% of all metastatic GC patients) but the response to ICI is also observed in non-EBV, MSS subset. In addition, even within EBV-positive or MSI-H patient cohort, the response rate is 50% which suggests that there are other influential factors to determine response to immune checkpoint inhibitors. Of 13 non-EBV, MSS responders, 11 (86%) patients had high chromatin openness in their circulating CD8⁺ T cells. Although definitive conclusion cannot be drawn from this study due to small sample size, this is the first study to demonstrate that circulating T-cell characteristics may contribute in predicting response to ICI. Since this study was not designed as a robust biomarker validation study, we are currently designing a biomarker validation study in MSS metastatic GC patients undergoing ICI therapy to evaluate if pre-therapy T cell epigenetics can predict response beyond PD-L1. Thus, we have improved the result and discussion section in the revised manuscript accordingly (*page 13 lines 4-5; page 15 lines 5-17*)

Second, we agree with your comment that the first cohort was enriched in EBV/MSI-H responders. To overcome this shortness, we included MSS/EBV- GC patients in the validation cohort ($n = 52$) with 18 responders of whom 13 are EBV negative and MSS (but responders). Although it is preliminary, our assay showed that 11 of 13 patients had high openness in CD8⁺ T cells in MSS/EBV- GC. Lastly, in order to explicitly demonstrate each patient's characteristics, we provided each patient's demographic and pathologic data in Supplementary Table S1 and S2 in the revised manuscript.

[Redacted]

[Redacted]

Major comments:

1. The authors should describe in detail their experimental ATAC-seq protocol, and quality control measurements for each sample. QC statistics of interest are TSS scores and fraction reads in peaks metrics for each sample. Sample quality filtering is not described at all in the results section- did all samples pass TSS/FRIP score filtering? How many cells were assayed in each sample?

(Authors' Response/Action) We appreciate the reviewer for bring this to our attention. We tried to revised ANALYTICAL METHODS as the reviewer's comments. In revised Supplementary Table S3, statistical information with fraction of reads in peaks (FRiP) and transcription start site (TSS) enrichments were included as below.

FRiP scores and TSS enrichment score of all samples in discovery cohorts show greater than 0.3 and 10 as statistically preferred, respectively. These calculations are described in the section of ATAC-Seq quality control: Transcription start site (TSS) enrichments and fraction of reads in peaks from ANALYTICAL METHODS in the revised Supplementary Material.

Table S3. Assay for transposase-accessible chromatin using sequencing (ATAC-seq) metadata and quality control statistics.

Sample ID	Total reads	Aligned reads	% Aligned	% Duplicated	% Mitochondria	NRF	PBC	FRiP	TSS enrichment
SS01	79,369,131	72,958,765	91.9%	51.3%	31.4%	0.98	0.86	0.60	16.4
SS02	87,638,376	80,039,100	91.3%	43.0%	21.4%	0.98	0.83	0.47	16.0
SS03	126,685,260	115,837,870	91.4%	45.6%	21.2%	0.98	0.80	0.55	26.8
SS04	66,576,763	60,989,804	91.6%	43.1%	26.0%	0.98	0.89	0.64	28.2
SS05	69,794,733	63,072,918	90.4%	46.4%	26.2%	0.98	0.85	0.54	18.1
SS06	110,916,650	102,146,159	92.1%	46.1%	26.9%	0.98	0.86	0.53	11.9
SS07	87,534,113	80,315,836	91.8%	50.5%	29.0%	0.98	0.83	0.50	14.2
SS08	79,366,311	73,459,580	92.6%	41.6%	23.9%	0.98	0.88	0.53	15.7
SS09	111,528,381	102,440,329	91.9%	47.7%	21.2%	0.98	0.77	0.51	16.9
SS10	106,022,192	97,579,178	92.0%	46.6%	24.4%	0.98	0.83	0.51	21.3
SS11	71,755,642	66,493,486	92.7%	38.3%	20.8%	0.98	0.88	0.46	16.3
SS12	88,065,358	82,845,759	94.1%	47.2%	24.7%	0.98	0.82	0.42	14.3
SS13	64,060,244	60,505,431	94.5%	44.3%	21.5%	0.97	0.83	0.44	19.2
SS14	103,257,981	97,746,391	94.7%	39.3%	13.3%	0.98	0.76	0.51	16.8
SS15	124,158,216	116,490,224	93.8%	53.5%	16.8%	0.97	0.66	0.57	16.3
SS16	73,535,141	69,458,094	94.5%	36.6%	18.5%	0.98	0.87	0.53	21.9
SS17	84,446,831	79,526,537	94.2%	38.0%	14.7%	0.98	0.80	0.38	16.3
SS18	61,138,757	58,835,255	96.2%	36.3%	18.9%	0.98	0.87	0.49	26.6
SS19	74,499,318	71,572,361	96.1%	44.2%	21.4%	0.98	0.81	0.41	26.2
SS20	80,080,760	76,640,704	95.7%	41.4%	17.6%	0.98	0.80	0.48	16.1
SS21	70,826,527	67,917,053	95.9%	40.5%	18.0%	0.98	0.82	0.54	17.2
SS22	89,838,238	86,206,816	96.0%	45.4%	22.9%	0.97	0.83	0.48	14.0
SS23	75,306,813	69,825,407	92.7%	37.3%	18.0%	0.98	0.86	0.35	10.8
SS24	115,656,817	107,321,051	92.8%	39.1%	14.7%	0.98	0.79	0.37	12.3
SS25	89,273,685	82,918,784	92.9%	49.1%	24.5%	0.98	0.79	0.42	15.3
SS26	107,117,518	99,277,606	92.7%	32.6%	11.6%	0.98	0.84	0.48	13.4
SS27	52,636,072	48,671,987	92.5%	38.5%	20.4%	0.98	0.87	0.44	15.8
SS28	64,582,832	60,171,173	93.2%	43.7%	23.8%	0.98	0.86	0.42	14.8
SS29	80,226,651	75,580,819	94.2%	57.0%	33.6%	0.97	0.81	0.46	23.0
SS30	101,999,249	95,839,289	94.0%	54.0%	30.3%	0.97	0.81	0.39	13.5
SS31	49,007,843	45,924,143	93.7%	30.1%	14.7%	0.98	0.89	0.61	24.0
SS32	114,145,932	107,358,491	94.1%	53.5%	24.7%	0.97	0.74	0.54	20.5

2. The step 2 filtering process in Fig 3A is quite difficult to understand. The authors should carefully describe what exactly is done to go from 121 to 67 peaks, and give examples of features of the peaks that are filtered out at this stage.

(Authors' Response/Action) We deeply appreciate the reviewer for this comment. To avoid confusion, we have revised the sentence in the revised manuscript accordingly (*page 11 lines 9-13; page 20 lines 16-18*).

We explained this clearly as follows: To quantitatively determine distinctive peaks between the responder and non-responder groups, area values of 2,560 peaks identified from three peak callers were normalized by multiplication with a normalization factor (F) calculated using three sets of controls, ranked -5 (C_5), -20 (C_{20}), and -50 (C_{50}) (Supplementary Table S6). Selected 68, 121 and 149 peaks by three sets of controls respectively showed the normalized average area value of the cohort greater than 10,000 and the mean difference in area between the responder and non-responder groups statistically significant ($p < 0.05$, Mann-Whitney U test). Only 67 peaks among normalized peaks by three sets of controls satisfied these selection criteria (Figure 3A and Supplementary Figure S5).

Figure 3A

Supplementary Figure S5

3. A major issue is the progression from 67 peaks discovered in the unbiased manner described in Fig 3A to 9 peaks described in Figures 4 and 5. Why were 9 peaks chosen and not 10 or 11? What happens to the performance (particularly specificity) of the test if all 67 peaks are included? How do all 67 peaks perform in the validation cohort? The authors should provide the sensitivity/specificity data for each of the 67 peaks that were identified.

(Authors' Response/Action) We appreciate the reviewer for this comment. As suggested by the reviewer, we revised the manuscript including ANALYTICAL METHODS and figures (*page 11 lines 9-13; page 20 lines 16-18*).

To clearly define 9 representative peaks among 67 peaks, three selection criteria were used. First, 3 peaks were selected with higher average in the responder group. Another 3 peaks were selected with lower variance among area values in the responder group. The other 3 peaks with highest relative distance between two groups were selected. This relative distance was calculated as follows: The area value of each peak was divided by the largest area value among all samples, and then the relative distance was calculated with the average of each group.

As the reviewer's recommendation, we analyzed total 67 peaks in our combination set for the diagnostic performance. The sensitivity and the specificity in the discovery cohort were reached to 100% at the combination with 6 peaks and 95.5% at the combination with 7 peaks, suggesting the combination with 9 peaks could be optimal for the performance.

Lastly, as suggested by the reviewer, we have updated Supplementary Figs S8 and S10 by adding the sensitivity/specificity data for each of the 67 peaks that were identified in the discovery and validation cohorts.

Supplementary Figure S8

A

Target ID	AUROC	Sample (n)		Sensitivity	Specificity
		Threshold	NR**		
M1	0.745	> 7 ≤ 3	8 14	70.0%	63.6%
M2	0.777	> 9 ≤ 1	8 14	90.0%	63.6%
M3	0.886	> 9 ≤ 1	4 18	90.0%	81.8%
M4	0.718	> 7 ≤ 3	7 15	70.0%	68.2%
M5	0.841	> 9 ≤ 1	7 15	90.0%	68.2%
M6	0.768	> 7 ≤ 3	5 17	70.0%	77.3%
M7	0.845	> 9 ≤ 1	7 15	90.0%	68.2%
M8	0.759	> 8 ≤ 2	4 18	80.0%	81.8%
M9	0.761	> 8 ≤ 2	13 9	80.0%	40.9%
M10	0.718	> 9 ≤ 1	10 12	90.0%	54.5%
M11	0.786	> 9 ≤ 1	11 11	90.0%	50.0%
M12	0.705	> 9 ≤ 1	12 10	90.0%	45.5%
M13	0.809	> 8 ≤ 2	5 17	80.0%	77.3%
M14	0.759	> 9 ≤ 1	8 14	90.0%	63.6%
M15	0.723	> 7 ≤ 3	9 13	70.0%	59.1%
M16	0.750	> 9 ≤ 1	8 14	90.0%	63.6%
M17	0.905	> 9 ≤ 1	4 18	90.0%	81.8%
M18	0.814	> 9 ≤ 1	8 14	90.0%	63.6%
M19	0.814	> 10 ≤ 0	7 15	100.0%	68.2%
M20	0.791	> 9 ≤ 1	6 16	90.0%	72.7%
M21	0.818	> 9 ≤ 1	6 16	90.0%	72.7%
M22	0.727	> 8 ≤ 2	9 13	80.0%	59.1%
M23	0.832	> 8 ≤ 2	5 17	80.0%	77.3%
M24	0.736	> 7 ≤ 3	7 15	70.0%	68.2%
M25	0.741	> 8 ≤ 2	8 14	80.0%	63.6%
M26	0.755	> 9 ≤ 1	9 13	90.0%	59.1%
M27	0.818	> 7 ≤ 3	5 17	70.0%	77.3%
M28	0.823	> 9 ≤ 1	8 14	90.0%	63.6%
M29	0.777	> 10 ≤ 0	9 13	100.0%	59.1%
M30	0.786	> 7 ≤ 3	5 17	70.0%	77.3%
M31	0.755	> 8 ≤ 2	8 14	80.0%	63.6%
M32	0.791	> 9 ≤ 1	9 13	90.0%	59.1%
M33	0.818	> 9 ≤ 1	9 13	90.0%	59.1%
M34	0.764	> 9 ≤ 1	8 14	90.0%	63.6%

Target ID	AUROC	Sample (n)		Sensitivity	Specificity
		Threshold	NR**		
M35	0.873	> 10 ≤ 0	11 11	100.0%	50.0%
M36	0.868	> 9 ≤ 1	4 18	90.0%	81.8%
M37	0.750	> 8 ≤ 2	5 17	80.0%	77.3%
M38	0.805	> 7 ≤ 3	6 16	70.0%	72.7%
M39	0.755	> 9 ≤ 1	7 15	90.0%	68.2%
M40	0.823	> 9 ≤ 1	5 17	90.0%	77.3%
M41	0.745	> 8 ≤ 2	8 14	80.0%	63.6%
M42	0.764	> 7 ≤ 3	5 17	70.0%	77.3%
M43	0.736	> 7 ≤ 3	6 16	70.0%	72.7%
M44	0.727	> 7 ≤ 3	4 18	70.0%	81.8%
M45	0.705	> 7 ≤ 3	6 16	70.0%	72.7%
M46	0.682	> 7 ≤ 3	9 13	70.0%	59.1%
M47	0.805	> 9 ≤ 1	7 15	90.0%	68.2%
M48	0.814	> 7 ≤ 3	3 19	70.0%	86.4%
M49	0.689	> 8 ≤ 2	8 14	80.0%	63.6%
M50	0.791	> 9 ≤ 1	9 13	90.0%	59.1%
M51	0.786	> 9 ≤ 1	7 15	90.0%	68.2%
M52	0.768	> 9 ≤ 1	9 13	90.0%	59.1%
M53	0.755	> 9 ≤ 1	9 13	90.0%	59.1%
M54	0.700	> 7 ≤ 3	6 16	70.0%	72.7%
M55	0.759	> 8 ≤ 2	7 15	80.0%	68.2%
M56	0.814	> 9 ≤ 1	7 15	90.0%	68.2%
M57	0.877	> 9 ≤ 1	4 18	90.0%	81.8%
M58	0.764	> 7 ≤ 3	5 17	70.0%	77.3%
M59	0.764	> 7 ≤ 3	4 18	70.0%	81.8%
M60	0.832	> 8 ≤ 2	4 18	80.0%	81.8%
M61	0.718	> 8 ≤ 2	7 15	80.0%	68.2%
M62	0.777	> 9 ≤ 1	7 15	90.0%	68.2%
M63	0.773	> 9 ≤ 1	8 14	90.0%	63.6%
M64	0.800	> 9 ≤ 1	6 16	90.0%	72.7%
M65	0.859	> 8 ≤ 2	3 19	80.0%	86.4%
M66	0.777	> 7 ≤ 3	5 17	70.0%	77.3%
M67	0.764	> 8 ≤ 2	6 16	80.0%	72.7%

R*, responder = CR + PR
NR**, non-responder = PD + SD

Supplementary Figure S10

Target ID	AUROC	Sample (n)		Sensitivity	Specificity
		Threshold	NR**		
M1	0.648	> 12	13	66.7%	61.8%
		≤ 6	21		
M2	0.703	> 12	11	66.7%	67.6%
		≤ 6	23		
M3	0.611	> 16	22	88.9%	35.3%
		≤ 2	12		
M4	0.660	> 16	20	88.9%	41.2%
		≤ 2	14		
M5	0.560	> 12	19	66.7%	44.1%
		≤ 6	15		
M6	0.389	> 7	19	38.9%	44.1%
		≤ 11	15		
M7	0.426	> 9	21	50.0%	38.2%
		≤ 9	13		
M8	0.467	> 8	18	44.4%	47.1%
		≤ 10	16		
M9	0.699	> 15	15	83.3%	55.9%
		≤ 3	19		
M10	0.529	> 15	20	83.3%	41.2%
		≤ 3	14		
M11	0.694	> 14	19	77.8%	44.1%
		≤ 4	15		
M12	0.709	> 12	13	66.7%	61.8%
		≤ 6	21		
M13	0.645	> 13	20	72.2%	41.2%
		≤ 5	14		
M14	0.665	> 13	21	72.2%	38.2%
		≤ 5	13		
M15	0.547	> 12	19	66.7%	44.1%
		≤ 6	15		
M16	0.482	> 13	24	72.2%	29.4%
		≤ 5	10		
M17	0.631	> 16	21	88.9%	38.2%
		≤ 2	13		
M18	0.614	> 15	20	83.3%	41.2%
		≤ 3	14		
M19	0.668	> 13	15	72.2%	55.9%
		≤ 5	19		
M20	0.712	> 17	20	94.4%	41.2%
		≤ 1	14		
M21	0.565	> 12	20	66.7%	41.2%
		≤ 6	14		
M22	0.670	> 16	17	88.9%	50.0%
		≤ 2	17		
M23	0.538	> 14	24	77.8%	29.4%
		≤ 4	10		
M24	0.644	> 12	16	66.7%	52.9%
		≤ 6	18		
M25	0.668	> 13	18	72.2%	47.1%
		≤ 5	16		
M26	0.663	> 16	18	88.9%	47.1%
		≤ 2	16		
M27	0.575	> 13	21	72.2%	38.2%
		≤ 5	13		
M28	0.605	> 11	16	61.1%	52.9%
		≤ 7	18		
M29	0.740	> 16	19	88.9%	44.1%
		≤ 2	15		
M30	0.618	> 15	18	83.3%	47.1%
		≤ 3	16		
M31	0.641	> 14	19	77.8%	44.1%
		≤ 4	15		
M32	0.621	> 10	15	55.6%	55.9%
		≤ 8	19		
M33	0.804	> 13	9	72.2%	73.5%
		≤ 5	25		
M34	0.681	> 14	18	77.8%	47.1%
		≤ 4	18		

Target ID	AUROC	Sample (n)		Sensitivity	Specificity
		Threshold	NR**		
M35	0.544	> 16	25	88.9%	26.5%
		≤ 2	9		
M36	0.619	> 14	17	77.8%	50.0%
		≤ 4	17		
M37	0.660	> 12	16	66.7%	52.9%
		≤ 6	18		
M38	0.592	> 14	19	77.8%	44.1%
		≤ 4	15		
M39	0.596	> 16	22	88.9%	35.3%
		≤ 2	12		
M40	0.637	> 14	22	77.8%	35.3%
		≤ 4	12		
M41	0.508	> 13	25	72.2%	26.5%
		≤ 5	9		
M42	0.616	> 8	13	44.4%	61.8%
		≤ 10	21		
M43	0.617	> 13	17	72.2%	50.0%
		≤ 5	17		
M44	0.588	> 8	14	44.4%	58.8%
		≤ 10	20		
M45	0.636	> 15	21	83.3%	38.2%
		≤ 13	13		
M46	0.605	> 11	15	61.1%	55.9%
		≤ 7	19		
M47	0.654	> 12	17	66.7%	50.0%
		≤ 6	15		
M48	0.541	> 13	23	72.2%	32.4%
		≤ 5	11		
M49	0.516	> 12	22	66.7%	35.3%
		≤ 6	12		
M50	0.649	> 14	18	77.8%	47.1%
		≤ 4	16		
M51	0.471	> 14	25	77.8%	26.5%
		≤ 4	9		
M52	0.511	> 15	22	83.3%	35.3%
		≤ 3	12		
M53	0.613	> 12	19	66.7%	44.1%
		≤ 6	15		
M54	0.626	> 14	20	77.8%	41.2%
		≤ 4	14		
M55	0.694	> 18	21	100.0%	38.2%
		≤ 0	13		
M56	0.626	> 16	27	89.9%	20.6%
		≤ 2	7		
M57	0.745	> 10	11	56.6%	67.6%
		≤ 8	23		
M58	0.560	> 14	23	77.8%	32.4%
		≤ 4	11		
M59	0.725	> 12	12	66.7%	64.7%
		≤ 6	22		
M60	0.688	> 17	23	94.4%	32.4%
		≤ 1	11		
M61	0.513	> 13	24	72.2%	29.4%
		≤ 5	10		
M62	0.546	> 16	23	89.9%	32.4%
		≤ 2	11		
M63	0.665	> 13	17	72.2%	50.0%
		≤ 5	17		
M64	0.641	> 12	17	66.7%	50.0%
		≤ 6	17		
M65	0.717	> 15	16	83.3%	52.9%
		≤ 3	18		
M66	0.624	> 15	24	83.3%	29.4%
		≤ 3	10		
M67	0.755	> 16	16	89.9%	52.9%
		≤ 2	16		

R*, responder = CR + PR
 NR**, non-responder = PD + SD

4. The authors should describe the identity of the 67 peaks in more detail. I might suggest a GREAT analysis (great.stanford.edu) of nearby genes, enriched pathways, etc. Further, are these peaks memory or naïve T cell specific (or common peaks)? Are they open in other cell types? The authors could compare to some of the reference datasets that they describe earlier in the manuscript.

(Authors' Response/Action) We appreciate the reviewer's comment. To define the identity of the 67 peaks, these peaks were compared to ATAC-seq datasets (GSE89308) from PB CD8⁺ T-cell subsets as described in our manuscript. Each peak area of PB CD8⁺ T-cell subsets was normalized as a same manner described in the method section and displayed in a stacked bar graph to proportionally compare the chromatin openness of 67 peaks among naïve, effector memory (EM) and central memory (CM) CD8⁺ T cells. While most peaks were opened in naïve CD8⁺ T cells, peaks highly opened in CM CD8⁺ T cells and to a lesser extent, in EM CD8⁺ T cells. Thus, the fact that the selected 67 peaks are well opened in memory CD8⁺ T cells may well reflect our hypothesis that chronic stress induces epigenetic changes and leaves traces in the immune system.

Further, to address if these peaks open in other cell types, we analyzed ATAC-seq data of normal bronchial epithelial cells, small cell lung cancer cells, normal prostate basal epithelial cells, and small cell prostate cancer cells from GSE118204. Each sample were performed “findPeaks.pl in Homer suite with default parameters” to identify opened regions and these regions were compared with 67 peaks using “findOverlaps” function in “GenomicRanges” package. The opened region overlapped with the 67 peaks were displayed in filled circle, the others were displayed in empty circle. According to this analysis, although 67 peaks in immune cells except monocytes open about 70% (*data now shown*), peaks in other cell types mentioned above only opened less than 50%. These peak regions might be immune-cell specific, related to immune regulation.

As the reviewer's comment, we tried to define any functional correlation to known biological pathway. Only 31 peaks located within a gene body or putative regulatory regions ranged between from TSS to -30 kb were considered. Using these peaks annotated as a gene nearest or within, we tried to discover the functional features of gene regulation. From Ingenuity Pathway Analysis program, genes were mostly categorized in homeostasis, differentiation, movement, proliferation, anergy of T lymphocytes. We wish this experiment can be pursued in the follow-up paper that we are working on now.

5. The discussion regarding plastic dysfunctional cells is provocative, but has no support from the presented data. The fact that AP-1 or NFAT motifs are present in the differential peaks may represent some relation to T cell dysfunction, or more likely, just the fact that they are indeed T cells where these TFs are common and active. In other words, AP-1 and NFAT are active in normal T cell development and activation, and in exhaustion, and there is no way to know if the presence of binding sites in a peak represents an activation or exhaustion signature. Are the 67 peaks enriched in dysfunction or exhaustion-associated genes or TFs?

(Authors' Response/Action) We really agree to the reviewer's comment whether this study is more likely to discover biomarker, reflecting the benefit of anti-PD-1 therapy. As the reviewer mentioned, there is no way to explain how transcription factors identified by motif regulate genes associated with any cellular function. Based on our pathway analysis, immune-regulatory categories were found. Please see our response to the comment #4. Only two genes were selected for the anergy of T lymphocytes category, which is believed to be associated with exhaustion/dysfunction of T cells.

In particular, the authors make a broad statement on line 323 that 'These findings suggest that chromatin openness of specific genomic positions in exhausted CD8+ T cells is probably associated with the

restoration of T-cell functions by PD-1 blockade.’ This statement is not presently supported by any of the data in the manuscript and should be removed.

(Authors’ Response/Action) As suggested by the reviewer, we have deleted this statement in the revised manuscript (*page 15 line 5*).

6. Finally, there is a concern that the data is confounded by EBV and MSI status. For example, if I examine responders who are EBV/MSI negative, the sensitivity and specificity appear to go down. In Fig 5C, only 2/5 high openness patients that are EBV/MSI- respond to therapy. Is the present peak set and chromatin analysis applicable to all GC patients, or only those that are EBV/MSI +? The authors should run an analysis on EBV/MSI- patients and report sensitivity and specificity in both cohorts.

(Authors’ Response/Action) We appreciate the reviewer for bring this to our attention. We tried to address this question as raised by the reviewer #1. Please refer to a response for the comment of reviewer #1.

Reviewer #3 (Remarks to the Author):

In this manuscript, the relationship between epigenetic characteristics of baseline CD8⁺ T cells and anti-PD1 treatment is considered. A clinical trial (NCT#02589496) of 32 mGC patients is considered as the discovery cohort and another clinical trial of 22 patients who received pembrolizumab is considered as the validation cohort. However, there are no any biographical data of patients except patients' age. I have the following questions:

(Authors' Response/Action) We are grateful for your kind comments and have tried our best to revise the manuscript accordingly.

1. Are the characteristics of baseline CD8⁺ T cells relative to patient biographical variables?

(Authors' Response/Action) At the beginning of this study, we tried to find out the association with response to anti-PD-1 therapy, focusing on the frequency of CD8⁺ T cells and PD-1 expression of CD8⁺ T cells in 35 patients. In the flow cytometric analysis, there was no significant difference in the frequencies of CD8⁺ T cells and PD-1⁺CD8⁺ T cells in responders (complete response [CR], partial response [PR]) or non-responders (stable disease [SD] or progressive disease [PD]) to pembrolizumab (Fig. 1A and B). In addition, there was no substantial difference in percentage of Ki-67⁺ in PD-1⁺CD8⁺ T cells among responders and non-responders (Fig. 1C, right panel); this finding is consistent with the results of previous studies. Thus, we did not characterize the CD8⁺ T cells of samples in the validation cohort because we thought it was not meaningful to perform further analysis. Unfortunately, the data on this is nothing more than what is presented as there are no samples left. Please understand this situation.

2. As a biomarker, the cut-of-points in the validation cohort should be the same as that in the discovery cohort. However, the thresholds in the discovery cohort (Figure 4B) are different with the thresholds in the validation cohort (Figure 5B). It is also in Figure 4D (the cut-of-point =28.5) and in Figure 5D (the cut-of-point = 26.5).

(Authors' Response/Action) We deeply appreciate the reviewer for bring this to our attention. Thus, we carefully reviewed our data and re-analyzed all ATAC-seq data for the following reason. In preparing ATAC-seq libraries, the sample should be amplified using a few PCR cycles as possible. This might help to reduce PCR duplicates, which are exact copies of DNA fragments that can interfere with the biological signal of interest. During the analysis process, we could remove PCR duplicates. Based on these, we tried to re-analyzed all ATAC-seq data in the discovery and the validation cohort and updated the data in the revised manuscript.

Importantly, as suggested by the reviewer, we have applied the same threshold values both in the discovery and validation cohorts and revised the data in the revised manuscript (*page 12, line2 22-23;*

Figure 5

B

Combination of Target (ID)	AUROC	Threshold	Sample (n)		Sensitivity	Specificity
			R*	NR**		
M36+M17	0.626	8.5	> 14	17	77.8%	50.0%
			≤ 4	17		
M36+M17+M20	0.688	15.5	> 14	13	77.8%	61.8%
			≤ 4	21		
M36+M17+M20+M30	0.730	15.5	> 18	18	100.0%	47.1%
			≤ 0	16		
M36+M17+M20+M30+M5	0.720	20.5	> 18	17	100.0%	50.0%
			≤ 0	17		
M36+M17+M20+M30+M5+M29	0.700	24.5	> 17	16	94.4%	52.9%
			≤ 1	18		
M36+M17+M20+M30+M5+M29+M2	0.706	27.5	> 16	14	88.9%	58.8%
			≤ 2	20		
M36+M17+M20+M30+M5+M29+M2+M1	0.718	27.5	> 16	14	88.9%	58.8%
			≤ 2	20		
M36+M17+M20+M30+M5+M29+M2+M1+M35	0.717	28.5	> 16	14	88.9%	58.8%
			≤ 2	20		

R*, responder = CR + PR
NR**, non-responder = PD + SD

Supplementary Figure S10

Target ID	AUROC	Sample (n)		Sensitivity	Specificity
		Threshold	R* NR**		
M1	0.648	> 12 ≤ 6	13 21	66.7%	61.8%
M2	0.703	> 12 ≤ 6	11 23	66.7%	67.6%
M3	0.611	> 16 ≤ 2	22 12	88.9%	35.3%
M4	0.660	> 16 ≤ 2	20 14	88.9%	41.2%
M5	0.560	> 12 ≤ 6	19 15	66.7%	44.1%
M6	0.389	> 7 ≤ 11	19 15	38.9%	44.1%
M7	0.426	> 9 ≤ 9	21 13	50.0%	38.2%
M8	0.467	> 8 ≤ 10	18 16	44.4%	47.1%
M9	0.699	> 15 ≤ 3	15 19	83.3%	55.9%
M10	0.529	> 15 ≤ 3	20 14	83.3%	41.2%
M11	0.694	> 14 ≤ 4	19 15	77.8%	44.1%
M12	0.709	> 12 ≤ 6	13 21	66.7%	61.8%
M13	0.645	> 13 ≤ 5	20 14	72.2%	41.2%
M14	0.665	> 13 ≤ 5	21 13	72.2%	38.2%
M15	0.547	> 12 ≤ 6	19 15	66.7%	44.1%
M16	0.482	> 13 ≤ 5	24 10	72.2%	29.4%
M17	0.631	> 16 ≤ 2	21 13	88.9%	38.2%
M18	0.614	> 15 ≤ 3	20 14	83.3%	41.2%
M19	0.668	> 13 ≤ 5	15 19	72.2%	55.9%
M20	0.712	> 17 ≤ 1	20 14	94.4%	41.2%
M21	0.565	> 12 ≤ 6	20 14	66.7%	41.2%
M22	0.670	> 16 ≤ 2	17 17	88.9%	50.0%
M23	0.538	> 14 ≤ 4	24 10	77.8%	29.4%
M24	0.644	> 12 ≤ 6	16 18	66.7%	52.9%
M25	0.668	> 13 ≤ 5	18 16	72.2%	47.1%
M26	0.663	> 16 ≤ 2	18 16	88.9%	47.1%
M27	0.575	> 13 ≤ 5	21 13	72.2%	38.2%
M28	0.605	> 11 ≤ 7	16 18	61.1%	52.9%
M29	0.740	> 16 ≤ 2	19 15	88.9%	44.1%
M30	0.618	> 15 ≤ 3	18 16	83.3%	47.1%
M31	0.641	> 14 ≤ 4	19 15	77.8%	44.1%
M32	0.621	> 10 ≤ 8	15 19	55.6%	55.9%
M33	0.804	> 13 ≤ 5	9 25	72.2%	73.5%
M34	0.681	> 14 ≤ 4	18 18	77.8%	47.1%
M35	0.544	> 16 ≤ 2	25 9	88.9%	26.5%
M36	0.619	> 14 ≤ 4	17 17	77.8%	50.0%
M37	0.660	> 12 ≤ 6	16 18	66.7%	52.9%
M38	0.592	> 14 ≤ 4	19 15	77.8%	44.1%
M39	0.596	> 16 ≤ 2	22 12	88.9%	35.3%
M40	0.637	> 14 ≤ 4	22 12	77.8%	35.3%
M41	0.508	> 13 ≤ 5	25 9	72.2%	26.5%
M42	0.616	> 8 ≤ 10	13 21	44.4%	61.8%
M43	0.617	> 13 ≤ 5	17 17	72.2%	50.0%
M44	0.588	> 8 ≤ 10	14 20	44.4%	58.8%
M45	0.636	> 15 ≤ 13	21 13	83.3%	38.2%
M46	0.605	> 11 ≤ 7	15 19	61.1%	55.9%
M47	0.654	> 12 ≤ 6	17 15	66.7%	50.0%
M48	0.541	> 13 ≤ 5	23 11	72.2%	32.4%
M49	0.516	> 12 ≤ 6	22 12	66.7%	35.3%
M50	0.649	> 14 ≤ 4	18 16	77.8%	47.1%
M51	0.471	> 14 ≤ 4	25 9	77.8%	26.5%
M52	0.511	> 15 ≤ 3	22 12	83.3%	35.3%
M53	0.613	> 12 ≤ 6	19 15	66.7%	44.1%
M54	0.626	> 14 ≤ 4	20 14	77.8%	41.2%
M55	0.694	> 18 ≤ 0	21 13	100.0%	38.2%
M56	0.626	> 16 ≤ 2	27 7	89.9%	20.6%
M57	0.745	> 10 ≤ 8	11 23	56.6%	67.6%
M58	0.560	> 14 ≤ 4	23 11	77.8%	32.4%
M59	0.725	> 12 ≤ 6	12 22	66.7%	64.7%
M60	0.688	> 17 ≤ 1	23 11	94.4%	32.4%
M61	0.513	> 13 ≤ 5	24 10	72.2%	29.4%
M62	0.546	> 16 ≤ 2	23 11	89.9%	32.4%
M63	0.665	> 13 ≤ 5	17 17	72.2%	50.0%
M64	0.641	> 12 ≤ 6	17 17	66.7%	50.0%
M65	0.717	> 15 ≤ 3	16 18	83.3%	52.9%
M66	0.624	> 15 ≤ 3	24 10	83.3%	29.4%
M67	0.755	> 16 ≤ 2	16 16	89.9%	52.9%

R*, responder = CR + PR
NR**, non-responder = PD + SD

Supplementary Figure S11

3. The sample size in the validation cohort is not enough.

(Authors' Response/Action) To increase the sample size in the validation cohort as suggested by the reviewer, we tried to use additional baseline samples from the same validation cohort. We considered the overall size of the validation cohort based on Simon's two-stage optimal design, which was previously used in our study (Kim *et al.*, Nat Med 2018). The parameter used at this time was the true response rate was 35% with 90% power and to reject the hypothesis that the response rate was less than 15%, with a one-sided alpha of 5%. Thus, the validation cohort size is more than 44 patients with the same symptom and cohort size optimal for applying the same treatment in 52 patients in our new validation cohort, and it was judged to be a sufficient size to be applied in biomarker analysis. Finally, we updated the data in the revised manuscript (*page 12, line2 22-23; page 23 lines 1-14*) as described in the response to comment #2.

REVIEWER COMMENTS

Reviewer #1 (Remarks to the Author):

No further comments. The authors have adequately addressed all of my queries.

Signed,
Jun Gong, MD

Reviewer #3 (Remarks to the Author):

The manuscript has been great improved and all of my comments are addressed in this new version. I have no further comments.

Reviewer #4 (Remarks to the Author): to replace original Reviewer #2

The authors use samples from a clinical trial for metastatic gastric cancer patients treated with pembrolizumab to determine epigenetic markers in CD8 T cells, which could be used to determine responders/non-responders. Since they use total CD8 T cells, the conceptual framework is difficult to understand. The epigenetic landscape in CD8 T cells is highly influenced by their differentiation state, in particular exhausted T cells have a different signature than normal effector T cells. Within exhausted T cells, there is small subset of T cells that has been described to be more responsive to PD1 blockade. It is therefore difficult to interpret the signature of such a mixed population. The transcription factor motif enrichment at differentially accessible sites that the authors find is more consistent with T cell activation (e.g. bZIP family, NFAT) than the signature of this subset (TCF). A discussion of these issues would be worthwhile.

Revisions have improved the manuscript.

1. Quality metrics of the ATAC-seq data are now presented
2. To analyze the ATACseq data, the authors devised a normalization method which identifies a set of "control" regions in the genome by using published ATAC and DNase I from ENCODE specifically at TSS regions which overlap with peaks from their study. From these regions they further filtered them for ones with less variability across samples. They further use these control regions to scale the peak intensities within their samples to identify peaks differential across responders and non-responders. The overlap of differential sites after scaling with 5/20/50 control peaks with is 67 peaks were used for prediction of response. The approach is now described in more detail. The authors used a cutoff of average area values > 10,000 to arrive at a differential peak set, however they don't specify how they decide this cutoff. It appears to be an arbitrary threshold.
3. From these the authors used ROC to determine 9 sites that provide "fair" diagnostic ability. They further show them as predictive for progression free survival in discovery and validation cohorts. In the discovery cohort, Fig. 4B combination of 8 sites provide identical results at lower threshold compared to 9 sites. The authors don't provide any rationale to choosing 9 over 8 sites for further analysis. It is apparent that a combination of a subset of these peaks are differentially accessible in different patients. The authors could perform an analysis to determine a cutoff for number of peaks from these set that should be differential in a patient to be predictive for progression free survival. The authors state that in context of validation cohort, "Each of the nine selected chromatin "open" regions predicted prolonged PFS in patients with mGC" however the p-values of 5 out of 9 genes (Supplementary Fig. S11) do not reach significance of < 0.05. This statement then would probably be misleading and authors do not provide explanation of this finding.
4. The authors now have now identified genes that are predicted to be regulated by differentially accessible sites. It would be of interested to see the mean tracks of accessibility of these sites in the different cohorts. In particular, PRDM1 is a transcription factor of interest in T cell

differentiation.

5. The authors emphasize sites with increased accessibility throughout the manuscript. What about sites with reduced accessibility.

Minor comments

1. Figure 1 Legend has a typo, description of B is labeled as A.
2. The text description labeled as Fig. 4C pertains to Fig. 4D (a typo? Figure-4C is then discussed in the text.)
3. The authors state that they use "Illumina NextSeq500 for 75 single-read bases", I wonder if this means they used single-end instead of pair-end sequencing. They mention they provided detailed protocol in supplement, however I did not find any methods in the attached supplement file. If they do use single-end sequencing, then this would also be a caveat as we use "Insert size distribution" from paired-end sequencing as a QC metric to assess the quality of samples and it also informs the peak size determinations. This study probably then be much more sensitive to peak sizes as they average the "openness area" across the different control peaks.
4. When the authors discuss transcription factor binding, they actually discuss transcription factor motif enrichment.

Responses to Reviewers' Comments

(Authors' Response/Action) We appreciate your comments very much. We revised the manuscript in accordance with your advice. In the following, we addressed the reviewers' comments point-by-point and made other necessary corrections. The revision made to this manuscript are marked in yellow-highlights.

REVIEWER COMMENTS

Reviewer #1 (Remarks to the Author):

No further comments. The authors have adequately addressed all of my queries.

Signed,

Jun Gong, MD

Reviewer #3 (Remarks to the Author):

The manuscript has been great improved and all of my comments are addressed in this new version. I have no further comments.

Reviewer #4 (Remarks to the Author): to replace original Reviewer #2

The authors use samples from a clinical trial for metastatic gastric cancer patients treated with pembrolizumab to determine epigenetic markers in CD8 T cells, which could be used to determine responders/non-responders. Since they use total CD8 T cells, the conceptual framework is difficult to understand. The epigenetic landscape in CD8 T cells is highly influenced by their differentiation state, in particular exhausted T cells have a different signature than normal effector T cells. Within exhausted T cells, there is small subset of T cells that has been described to be more responsive to PD1 blockade. It is therefore difficult to interpret the signature of such a mixed population. The transcription factor motif enrichment at differentially accessible sites that the authors find is more consistent with T cell activation (e.g. bZIP family, NFAT) than the signature of this subset (TCF). A discussion of these issues would be worthwhile.

(Authors' Response/Action) We are grateful for your kind comments. We agree that small subsets of T cells might be more responsive to PD1 blockade. At the beginning of this study, we tried to find out the association with response to anti-PD-1 therapy, focusing on the frequency of CD8⁺ T cells and PD-1 expression of CD8⁺ T cells in 35 patients. In the flow cytometric analysis, there was no significant difference in the frequencies of CD8⁺ T cells and PD-1⁺CD8⁺ T cells in responders (complete response

[CR), partial response [PR]) or non-responders (stable disease [SD] or progressive disease [PD]) to pembrolizumab (Fig. 1A and B). In addition, there was no substantial difference in percentage of Ki-67⁺ in PD-1⁺CD8⁺ T cells among responders and non-responders (Fig. 1C, right panel); this finding is consistent with the results of previous studies. These data led us analyze the pool of circulating CD8⁺ T cells regardless of the proportion of T cell subsets. Without the further information of T cell subsets, our 67 targets in CD8⁺ T cells may be used to identify mGC patients who may benefit from anti-PD-1 antibody in addition to tumor profiling.

Revisions have improved the manuscript.

1. Quality metrics of the ATAC-seq data are now presented

2. To analyze the ATACseq data, the authors devised a normalization method which identifies a set of “control” regions in the genome by using published ATAC and DNase I from ENCODE specifically at TSS regions which overlap with peaks from their study. From these regions they further filtered them for ones with less variability across samples. They further use these control regions to scale the peak intensities within their samples to identify peaks differential across responders and non-responders. The overlap of differential sites after scaling with 5/20/50 control peaks with is 67 peaks were used for prediction of response. The approach is now described in more detail. The authors used a cutoff of average area values > 10,000 to arrive at a differential peak set, however they don’t specify how they decide this cutoff. It appears to be an arbitrary threshold.

(Authors’ Response/Action) We appreciate the reviewer for bring this to our attention. To simplify the cutoff process, we chose 10,000 average value as an arbitrary threshold, but as your comment, we revised this value to our original process, “Average area value of peaks > average area value of total peaks”. This cutoff process was applied to in main figure 3A and the revised manuscript (*page 11 line 11*) and supplementary materials (*page 18 line 25*).

3. From these the authors used ROC to determine 9 sites that provide “fair” diagnostic ability. They further show them as predictive for progression free survival in discovery and validation cohorts. In the discovery cohort, Fig. 4B combination of 8 sites provide identical results at lower threshold compared to 9 sites. The authors don’t provide any rationale to choosing 9 over 8 sites for further analysis. It is apparent that a combination of a subset of these peaks are differentially accessible in different patients. The authors could perform an analysis to determine a cutoff for number of peaks from these set that should be differential in a patient to be predictive for progression free survival.

(Authors' Response/Action) We appreciate the reviewer for this comment. In the previous our response from the comment of former reviewer, we tried to address why we choose 9 representative peaks among 67 peaks. As the former reviewer's recommendation, we analyzed total 67 peaks in our combination set for the diagnostic performance instead of calculating PFS with all peaks. The sensitivity and the specificity in the discovery cohort were reached to 100% at the combination with 6 peaks and 95.5% at the combination with 7 peaks, suggesting the combination with 9 peaks could be optimal for the performance.

The authors state that in context of validation cohort, "Each of the nine selected chromatin "open" regions predicted prolonged PFS in patients with mGC". however, the p-values of 5 out of 9 genes (Supplementary Fig. S11) do not reach significance of < 0.05 . This statement then would probably be misleading and authors do not provide explanation of this finding.

(Authors' Response/Action) We appreciate the reviewer for bring this to our attention. As reviewer's point-out, we have tried to our best to revise this statement in the revised manuscript (*page 13 lines 5-13*) accordingly.

"Four of the nine selected chromatin "open" regions showed statistically significant differences in PFS in the validation cohort, but nonetheless, each median PFS was greater in patients with high chromatin openness than in patients with closed chromatin openness. While the nine targets showed reasonable discriminative ability (AUROC >0.544) and target sensitivity and specificity of $80.2 \pm 3.5\%$ and $46.7 \pm 3.9\%$ (mean \pm SEM), respectively (Supplementary Fig. S10 and S11), the combination of these peaks significantly enhanced the discriminative ability in the validation cohort (AUROC 0.717) and sensitivity and specificity of this combination was reached to 88.9% and 58.8%, respectively (Fig. 5B).".

4. The authors now have now identified genes that are predicted to be regulated by differentially accessible sites. It would be of interested to see the mean tracks of accessibility of these sites in the different cohorts. In particular, PRDM1 is a transcription factor of interest in T cell differentiation.

(Authors' Response/Action) We appreciate the reviewer for this comment. As a representative, the mean tracks visualized 9 targets in discovery cohort and validation cohort for responders (R) and non-responders (NR) groups were generated. The mean enrichments for each hg19 coordinates in each group were displayed as heatmap in grayscale using IGV genome browser. The average peak area values of each group were indicated. The mean track visualization would provide the same information in the dot

plots with normalized area values (Supplementary Fig. S9 and S11, left plots). Thus, we did not include these mean tracks in main figures.

We corrected some mis-annotated gene symbol in Table S7, due to the annotation mistake in Homer suite. Unfortunately, gene symbol with PRDM1 in the peak was changed to ATG5. However, the data and information in this study has not changed.

5. The authors emphasize sites with increased accessibility throughout the manuscript. What about sites with reduced accessibility.

(Authors' Response/Action) We appreciate the reviewer for this comment. We could not see the peaks with reduced accessibility only in a responder group. Most of differential peaks were found in a responder group, but not in a non-responder group.

Minor comments

1. Figure 1 Legend has a typo, description of B is labeled as A. We corrected this in the figure legend.
2. The text description labeled as Fig. 4C pertains to Fig. 4D (a typo? Figure-4C is then discussed in the text.) We corrected this in the revised manuscript (Page 12 lines 16-17)
3. The authors state that they use “Illumina NextSeq500 for 75 single-read bases”, I wonder if this means they used single-end instead of pair-end sequencing. They mention they provided detailed protocol in supplement, however I did not find any methods in the attached supplement file. If they do use single-end sequencing, then this would also be a caveat as we use “Insert size distribution” from paired-end sequencing as a QC metric to assess the quality of samples and it also informs the peak size determinations. This study probably then be much more sensitive to peak sizes as they average the “openness area” across the different control peaks.

(Authors’ Response/Action) We appreciate the reviewer for this comment. We carefully checked the details in ANALYTICAL METHODS and this wrong statement could mislead reader for the process of ATAC-seq. We have revised this statement in the revised manuscript (page 7 lines 18-19): Detailed method for ATAC-seq is provided in supplementary material. → Detailed further analytical method for ATAC-seq is provided in supplementary material.

We appreciate the reviewer for bring this peak size issue to our attention. We did not assess “Insert size distribution” from paired-end sequencing, due to the our FASTQ data from single read sequencing. However, we followed the guideline of ATAC library and deep-sequencing quality provided by Harvard FAS informatics and ENCODE consortium and data sets in this study meet the requirement of this quality guidelines (Supplementary Table S3). After peak calling process, peak size might be different between single read sequencing and paired-end sequencing as mentioned (bioRxiv, doi: <https://doi.org/10.1101/496521>). Before we decided single read sequencing instead of paired-end sequencing, 4 samples used in this study were generated to paired-end libraries and libraries were sequenced using Illumina NextSeq 500 system using 75-bp single read and Illumina NovaSeq 6000 system using 150-bp paired reads. We followed analytical process using MACS2 v2.1.2 and the same option, suggested from the preprint version of manuscript (bioRxiv, doi: <https://doi.org/10.1101/496521>). Then after peak-calling with MACS2, we compared 147bp fragment size of single read sequencing and average fragment size of paired-end sequencing.

Sample	# Raw fragments	# Filtered aligned fragments	Average size of fragments (bp)
--------	-----------------	------------------------------	--------------------------------

A	51,973,669	39,919,597	155.2
B	57,525,277	35,568,830	165.9
C	53,279,622	37,438,860	150.8
D	72,465,762	50,563,981	142.9

The mean and standard error of mean (SEM) of average size of fragments is 153.7 ± 4.15 bp, and the average size of fragment from pooled sample is 142.9 bp using 163,491,268 fragments. Therefore, the assumption that the fragment size as 147 bp for peak calling in single reads could be similar to paired read sequencing at least in our dataset.

Peak size comparison,

Venn diagram showing the overlap of peaks identified in both single reads and paired reads (note; Pooled sample ($n=4$), MACS2 with P -value <0.05)

Peak range distribution of 232 controls and 67 markers in both single reads and paired reads (note; Pooled sample ($n=4$), MACS2 with P -value <0.05)

As reviewer's comments, we expected that our single read sequencing is possibly sensitive to peak size in our data set, compared to paired-end sequencing. However, although we have the comparison with small size of single read and paired end sequencing dataset, we could not find critical difference in this comparison. We understand this set cannot reflect our entire dataset in this study. Please consider our case.

4. When the authors discuss transcription factor binding, they actually discuss transcription factor motif enrichment.

As suggested by the reviewer, we revised this in the revised manuscript (*page 15, line 8-9*)

REVIEWERS' COMMENTS

Reviewer #4 (Remarks to the Author):

The authors have addressed my comments.